# Airborne validation of radiative transfer modelling of ice clouds at millimetre and sub-millimetre wavelengths

Stuart Fox[1], Jana Mendrok[2], Patrick Eriksson[2], Robin Ekelund[2], Sebastian J. O'Shea[3], Keith N. Bower[3], Anthony J. Baran[1,4], R. Chawn Harlow[1], and Juliet C. Pickering[5]

[1]Met Office, FitzRoy Road, Exeter, UK, EX1 3PB
[2]Department of Space, Earth and Environment, Chalmers University of Technology, 41296 Gothenburg, Sweden
[3]Centre for Atmospheric Science, School of Earth and Environmental Science, University of Manchester, Manchester, UK, M13 9PL
[4]School of Physics, Astronomy and Mathematics, University of Hertfordshire, Hatfield, UK, AL10 9AB
[5]Space and Atmospheric Physics Group, Imperial College London, London, UK

*Correspondence to:* S. Fox (stuart.fox@metoffice.gov.uk)

**Abstract.** The next generation of European polar orbiting weather satellites will carry a novel instrument, the Ice Cloud Imager (ICI), which uses passive observations between 183 and 664GHz to make daily global observations of cloud ice. Successful use of these observations requires accurate modelling of cloud ice scattering, and this study uses airborne observations from two flights of the Facility for Airborne Atmospheric Measurements (FAAM) BAe-146 research aircraft to validate radiative transfer simulations of cirrus clouds at frequencies between 325 and 664GHz using the Atmospheric Radiative Transfer Simulator (ARTS) and a state-of-the-art database of cloud ice optical properties. Particular care is taken to ensure that the inputs to the radiative transfer model are representative of the true atmospheric state by combining both remote-sensing and in-situ observations of the same clouds to create realistic vertical profiles of cloud properties that are consistent with both observed particle size distributions and bulk ice mass. The simulations are compared to measurements from the International Sub-Millimetre Airborne Radiometer (ISMAR), which is an airborne demonstrator for ICI. It is shown that whilst they are generally able to reproduce the observed cloud signals, for a given ice water path (IWP) there is considerable sensitivity to the cloud microphysics including the distribution of ice mass within the cloud and the ice particle habit. Accurate retrievals from ICI will therefore require realistic representations of cloud microphysical properties.

# 1 Introduction

Ice cloud properties including column-integrated ice mass (ice water path, or IWP), particle size and cloud altitude can have an important impact on weather and climate prediction. However, they remain poorly constrained in current weather and climate models, in part due to the limited number of high-quality global observations (Waliser et al., 2009; Eliasson et al., 2013), and a recent study (Duncan and Eriksson, 2018) has shown that significant discrepancies still remain between both state-of-the art satellite datasets and reanalyses. Visible and infrared observations are mostly sensitive to particles close to the cloud tops, whilst current microwave observations, which are capable of penetrating deeper into the clouds, are restricted to frequencies up to 190GHz and are therefore relatively insensitive to all but the largest ice particles. Active sensing techniques, such as the combined lidar/radar DARDAR dataset (Delanoë and Hogan, 2010) are capable of sensing the full cloud depth with excellent vertical resolution, but they have very limited spatial coverage. Passive millimetre and sub-millimetre observations have been proposed for remote sensing of cloud ice properties by a number of authors (e.g. Buehler et al., 2007; Jiménez et al., 2007; Evans et al., 1998; Evans and Stephens, 1995a). Sub-millimetre waves are sensitive to scattering by cloud ice because the wavelengths are similar to the typical size of the cloud ice particles. However, with the exception of limb sounders designed for measuring atmospheric composition such as EOS MLS on the NASA's Aura satellite (Waters et al., 2006) and the Odin sub-millimetre radiometer (Murtagh et al., 2002) there are currently no satellite sub-millimetre radiometers capable of measuring cloud ice. This will change in the 2020s when the MetOp second generation satellites are launched, carrying the Ice Cloud Imager (ICI) (Kangas et al., 2012). This conical scanner with a 53°incidence angle will make passive observations between 183 and 664GHz to provide daily global coverage of cloud ice properties at 16km horizontal resolution. The retrieval of cloud ice properties from millimetre and sub-millimetre observations is a complex problem and it relies on accurate modelling of the interaction of the radiation with the cloud particles. However, validation of radiative transfer models at sub-millimetre frequencies is currently rather limited, due to the small number of available observations and the difficulty in obtaining co-located "ground truth" measurements.

Passive millimetre and sub-millimetre remote sensing of cloud ice relies on the scattering of warm upwelling radiation which is typically emitted by water vapour in the lower troposphere, leading to reduced brightness temperatures in the presence of ice clouds as described by Buehler et al. (2007). The magnitude and frequency dependence of the cloud-induced brightness temperature depressions depends on a number of parameters, including the ice water path, particle size distribution (PSD) and particle habit. Using multiple channels at different distances from the centre of a gaseous absorption line permits retrieval of the cloud altitude, and dual-polarised measurements can give an indication of particle orientation (Gong and Wu, 2017). A number of retrieval schemes have been developed and applied to existing millimetre and sub-millimetre observations. Evans et al. (2005, 2012) used a Bayesian Monte-Carlo approach to retrieve cloud ice properties from the CoSSIR instrument on board the NASA ER-2 aircraft during two campaigns. Retrievals of cloud ice have also been performed using measurements from the Odin-SMR and Aura MLS satellite limb-sounding radiometers (Eriksson et al., 2007; Rydberg et al., 2009; Wu et al., 2008). More recently Brath et al. (2018) have applied a neural-network based retrieval scheme to measurements from the ISMAR radiometer on the FAAM BAe-146 aircraft. A common feature of these retrievals is that they rely on databases or relationships

generated using radiative transfer models and make assumptions about the underlying ice microphysical properties. Whilst some validation of the retrieved products has been performed (e.g. Evans et al. (2005) compared retrieved integrated radar backscatter with the measurements from the 94GHz Cloud Radar System, and Eriksson et al. (2008) compared retrievals from Odin-SMR, Aura MLS and CloudSat) there has been little direct validation of the radiative transfer modelling. In part this is due to the difficulty of obtaining co-located measurements of millimetre and sub-millimetre brightness temperatures and the associated profiles of cloud microphysical properties required for input to the radiative transfer models. Airborne observations offer the potential to close this gap as they allow both remote-sensing and in-situ cloud microphysical measurements to be made for the same clouds. The aim of this paper is to validate the radiative transfer modelling of ice clouds at millimetre and sub-millimetre wavelengths by performing a closure study using observations from the FAAM BAe-146 atmospheric research aircraft.

The ISMAR radiometer (Fox et al., 2017) has been designed as an airborne demonstrator for ICI. It currently has channels at frequencies between 118 and 664GHz and flies on the FAAM aircraft along with the MARSS radiometer (McGrath and Hewison, 2001) which provides additional observations between 89 and 183GHz. In addition to the radiometers the FAAM aircraft can carry a large scientific payload. Of key interest here are the instruments which can be used to provide ground-truth measurements for input to radiative transfer models: a wide range of in-situ cloud microphysics probes measuring particle shapes, size distributions and bulk ice water content (IWC), and a downward-pointing 355nm Leosphere ALS-450 lidar which can measure cloud and aerosol profiles with high vertical and horizontal resolution. In this paper we will compare ISMAR/MARSS observations made during two cirrus flights with radiative transfer simulations. Inputs to the radiative transfer model will be derived from the aircraft observations in order to provide the best representation of the true atmospheric state at the time of the remote sensing observations. Even with an aircraft it is not possible to measure complete vertical profiles of cloud microphysical properties that are representative of the state of the cloud at the time and location of the remote sensing measurements due to the large spatio-temporal variability of typical clouds. Instead, we will combine remote sensing and in-situ measurements made at different times and locations to give the best estimate of the atmospheric state at the time and location of the ISMAR/MARSS observations. In particular we will derive profiles of cloud particle size distributions by combining volume extinction profiles from the ALS-450 lidar with empirical relationships derived from in-situ observations of the same clouds, and commonly used parametrizations. The particle habits used in the simulations will also be selected to be consistent with the in-situ observations.

The structure of the paper is as follows. Section 2 describes the flights used for the case studies, sec. 3 gives a brief description of aircraft instrumentation and sec. 4 describes the configuration of the ARTS radiative transfer model. The cloud microphysical measurements are described in sec. 5, and the comparison between the observations and radiative transfer simulations is presented in sec. 6. Finally, our conclusions are summarised in sec. 7.

## 2   Case studies

The measurements presented in this paper are taken from two FAAM flights targeting cirrus clouds. These two flights have been selected as they contain both above-cloud remote sensing observations and comprehensive in-situ characterisation of the

same clouds. Flight B895 took place on March 13th 2015 during the Cold-air Outbreak and Sub-Millimetre Ice Cloud Studies (COSMICS) campaign which ran in conjunction with the Cirrus Coupled Cloud-Radiation Experiment (CIRCCREX) project and was based in Prestwick, UK. It measured a relatively stationary, narrow, decaying band of cirrus cloud at altitudes between 6000m and 9000m associated with an occluded front. The temperatures at cloud base and top were approximately 245K and 221K respectively. The main part of the flight consisted of remote-sensing runs above the cloud band between 09:33UTC and 10:28UTC at an altitude of 9400m, followed by runs within the cloud between 10:33UTC and 11:28UTC at 8700m, 7900m, 7300m, 6700m and 6000m to gather in-situ measurements. Figure 1 shows the aircraft track during the remote sensing runs overlaid on a 10.8 μm infra-red satellite image from the 10:02UTC MetOp-B overpass. Further plots showing the aircraft tracks and satellite images during the in-cloud runs can be found in the supplement.

Flight B939 took place on 9th February 2016 during the Winter Experiment 2016 (WINTEX-16) campaign that was based in Cranfield, UK. The clouds were associated with a developing low pressure system that tracked eastwards from the English channel during the flight, and were at altitudes between 4000m and 7500m with temperatures at cloud base and top of approximately 252K and 224K respectively. The initial remote sensing runs took place above the clouds between 10:50UTC and 11:09UTC at an altitude of 8000m. These were followed by in-situ runs within the cloud between 11:12UTC and 11:54UTC at 7300m, 5700m and 4400m, and a run below the cloud at 3100m. This pattern of remote-sensing and in-situ runs was repeated during the flight, but in this paper we concentrate on the initial set of runs where clouds with the highest IWP were observed. Figure 1 shows a 10.8 μm infra-red satellite image from the 11:01UTC MetOp-A overpass that took place during the initial remote sensing runs, and plots showing satellite images and aircraft tracks during the in-cloud runs are available in the supplement. The western end of the track approached the edge of the clouds, and considerable variation in the clouds can be seen along the track.

## 3 Instrumentation

### 3.1 Passive millimetre and sub-millimetre observations

The passive millimetre and sub-millimetre observations were obtained using the MARSS and ISMAR radiometers on board the FAAM aircraft. MARSS was originally developed as an airborne demonstrator for AMSU-B, and ISMAR has been developed as an airborne demonstrator for ICI. In this section we give an overview of the two radiometers; full details of MARSS and ISMAR can be found in McGrath and Hewison (2001) and Fox et al. (2017).

Both radiometers contain a number of dual-sideband heterodyne receivers. They are mounted on the side of the aircraft and scan in the along-track direction allowing both upward and downward views. The available channels cover the range between 89GHz and 664GHz and include both atmospheric windows and $O_2$ and $H_2O$ absorption lines. In this paper we focus on the channels with centre frequencies at 325GHz and higher as the lower frequencies will have little sensitivity to cloud ice in cirrus clouds. However, as described in sec. 4, the 183GHz channels are used to constrain the atmospheric water vapour profile. A list of the channels used is given in tab. 1. In the downwards direction MARSS scans within a nominal +40°to -40° range, where positive angles refer to directions towards the front of the aircraft. However, in flight the combination of the aircraft

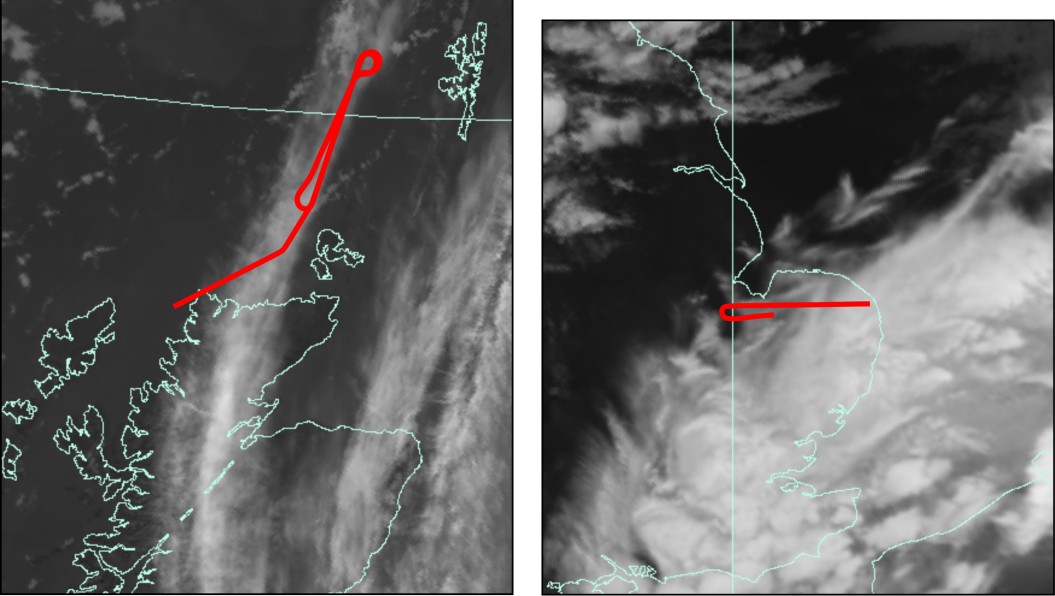

**Figure 1.** 10.8μm infra-red satellite image from 10:02UTC MetOp-B overpass on March 13th 2015 (left) and 11:01UTC MetOp-A overpass on February 9th 2016 (right). The aircraft track during the remote sensing runs is overlaid in red.

pitch angle and the angle at which the instrument is mounted on the aircraft give a maximum forward scan angle closer to +50°. The ISMAR scan range covers downwards angles in the nominal +50° to -10° range, and is mounted to the aircraft such that the nominal and actual scan angles are approximately the same during flight.

The majority of the MARSS and ISMAR receivers detect a single linear polarisation, whilst dual orthogonal polarisations are available in the window regions at 243GHz and 664GHz. The polarisation responses rotate as the instruments scan, and are designed such that they are close to V and H in the +50° downwards view in order to give the closest match to the conical scanning geometry used by ICI. The polarisation detected by each receiver is listed in tab. 1. However, in this study we will only use viewing directions close to nadir in order to achieve the closest match with the lidar viewing direction. All linear polarisations should be equivalent for nadir views.

## 3.2   Lidar

A Leosphere ALS-450 355nm elastic backscatter lidar was used to obtain cloud vertical profiles co-incident in space and time with the passive millimetre and sub-millimetre observations. The raw lidar profiles were measured with a vertical resolution of 1.5m and an integration time of 2s. Prior to further processing the profiles were smoothed to a vertical resolution of 45m and averaged over 10s, corresponding to a spatial distance of around 1.6-1.8km, to improve the signal-to-noise ratio. Cloud ice volume extinction profiles were retrieved from the range-corrected backscatter profiles using the two-stage process described by Marenco et al. (2011). In the first stage the lidar profiles which show a clear molecular scattering layer both above and

| Centre frequency (GHz) | Frequency offset (GHz) | IF Bandwidth (GHz) | Nadir NE$\Delta$T (K) | Polarisation (at +50° downwards view) | Feature | Instrument |
|---|---|---|---|---|---|---|
| 183.248 | ±0.975 | 0.45 | 0.5 | H | $H_2O$ | MARSS |
| | ±3.0 | 1.0 | 0.4 | | | |
| | ±7.0 | 2.0 | 0.3 | | | |
| 243.2 | ±2.5 | 3.0 | 0.3, 0.5 | H & V | Window | ISMAR |
| 325.15 | ±1.5 | 1.6 | 1.1 | V | $H_2O$ | ISMAR |
| | ±3.5 | 2.4 | 0.3 | | | |
| | ±9.5 | 3.0 | 0.8 | | | |
| 448.0 | ±1.4 | 1.2 | 0.9 | V | $H_2O$ | ISMAR |
| | ±3.0 | 2.0 | 1.3 | | | |
| | ±7.2 | 3.0 | 1.9 | | | |
| 664.0 | ±4.2 | 5.0 | 0.9, 2.7 | H & V | Window | ISMAR |

**Table 1.** ISMAR and MARSS channels with centre frequencies of 183GHz and higher. The column labelled NE$\Delta$T gives the channel noise-equivalent $\Delta$T, i.e. the brightness temperature difference that produces a change in signal equal to the radiometer noise level.

below the cloud are used to determine the average lidar ratio (extinction to backscatter ratio) for cloud ice particles using the method described by Di Girolamo et al. (1994). For B895 we found a mean value of $25\mathrm{sr}^{-1}$ and for B939 the mean value was $20.5\mathrm{sr}^{-1}$. These are close to the characteristic ice cloud lidar ratio for CALIPSO of $25\mathrm{sr}^{-1}$ given by Young et al. (2013).

This flight-average lidar ratio was then used in the standard Fernald-Klett inversion process (Fernald, 1984; Klett, 1985)
to derive the vertical profiles of particle extinction coefficient $\sigma$. For a stable inversion a molecular-scattering reference range below the cloud layer must be used which limits the retrieval to profiles where the lidar penetrates the full depth of the cloud layer. The maximum cloud optical depth that can be sensed depends on the performance of the lidar. Poor instrument alignment during flight B895 limited the maximum optical depth to 1, whereas cloud optical depths as high as 2 were observed during flight B939. The retrieved cloud extinction profiles are shown in fig. 2. Missing data correspond to aircraft turns or times when
the lidar did not penetrate the full depth of the cloud. There is considerable inhomogeneity in the cloud structure for both cases, with significant variability in the altitude and particle volume extinction in the different cloud layers. Higher peak values of extinction are observed for B939, but the vertical extent of the cloud is also generally smaller for this flight.

### 3.3 In-situ observations

Cloud particle images were collected using a number of probes in order to sample the full range of sizes that interact strongly
with millimetre and sub-millimetre radiation. On flight B895 the aircraft was fitted with a SPEC 2DS optical array probe (OAP)

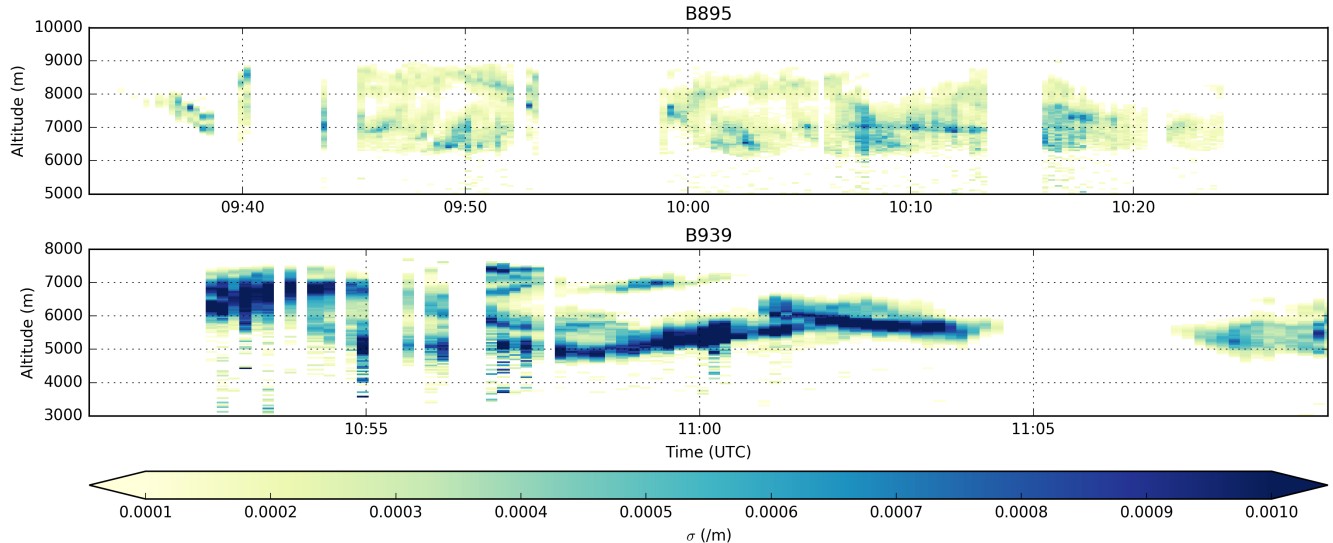

**Figure 2.** Lidar particle extinction profiles. Missing data correspond to aircraft turns or times when the lidar did not penetrate the full cloud depth.

which uses two orthogonal 128-element photodiode arrays to image particles in the nominal size range of 10-1280μm with a resolution of 10μm (Lawson et al., 2006). Larger particles (in the nominal size range of 100-6400μm) were imaged with 100μm resolution using a 64-element DMT CIP-100 optical array probe, and additional particle images were collected using a SPEC cloud particle imager (CPI), which uses a 1024×1024 pixel CCD camera to image particles with a 2.3μm resolution.

Artefacts caused by the shattering of large particles were reduced by using modified tips on the OAPs (Korolev et al., 2013a), and by applying an inter-arrival time threshold when processing the images (Field et al., 2006). Full details of the particle imaging instruments and processing for B895 can be found in O'Shea et al. (2016). On flight B939 the CPI and 2DS were not available so the smaller particles (nominal size range 15-960μm) were imaged with a resolution of 15μm using a 64-element DMT CIP-15 optical array probe.

Bulk ice water content was measured using a deep-cone Nevzorov hot-wire probe (Korolev et al., 2013b, 1998). The baseline correction described by Abel et al. (2014) was applied, which allows the probe to be sensitive to IWC as low as $0.002\mathrm{gm}^{-3}$. Ambient temperature was measured using a Rosemount/Goodrich-type 102 platinum resistance thermometer.

## 4 Radiative transfer model

The radiative transfer simulations have been performed using version 2.3 of the Atmospheric Radiative Transfer Simulator
(ARTS Eriksson et al., 2011; Buehler et al., 2018). This flexible code contains a number of modules that can be used to solve the scattering radiative transfer problem for different atmospheric and observation geometries, and degrees of polarisation. In this study we apply the RT4-version of PolRadTran (Evans and Stephens, 1991) as the scattering solver to calculate the

$I$ and $Q$ Stokes components of the radiation field separately for each individual observation along the aircraft flight track, assuming a 1-dimensional plane-parallel atmosphere. Note that for the near-nadir observations presented here the $Q$ component of the radiation field will be almost zero. The atmospheric properties for each simulation (e.g. vertical profiles of temperature, gas concentrations and hydrometeors) are constructed from a variety of sources including remote-sensing and in-situ aircraft observations and numerical weather prediction (NWP) data as described below.

Temperature and water vapour profiles are taken from the Met Office operational UKV NWP model (Tang et al., 2013), which has a spatial resolution across the UK of 1.5km. Profiles are available at hourly timesteps on 70 vertical levels between the surface and 50km. We have interpolated the profiles to the time and position of the aircraft observations using linear interpolation in time and nearest-neighbour interpolation in space. NWP model profiles are used rather than dropsonde profiles because they are capable of representing variability in the atmosphere at finer spatial resolution. Also, comparisons with radiative transfer simulations at 183GHz for clear-sky flights in similar geographical regions during the COSMICS and WINTEX-16 campaigns suggest that there is a significant dry bias in the dropsonde measurements. Nevertheless, a warm bias of 1-2K remained between the measurements and clear-sky simulations of brightness temperatures at 183GHz for the cirrus flights used in this study. At this frequency the scattering from cloud ice has a rather small impact, and would be expected to cause a cold bias when comparing observations to clear-sky simulations. This suggests that small errors still exist in the NWP water vapour profiles. We have therefore applied adjustments to the water vapour profiles by performing a 1D-VAR retrieval using the observations at 183±1 and 183±3GHz with the NWP profile as the background state and assuming clear-sky conditions.

Volume mixing ratios of $O_2$ and $N_2$ are fixed at 0.2095 and 0.7808, respectively, throughout the vertical profile. Oxygen and nitrogen absorption are calculated using the models of Tretyakov et al. (2005) and Rosenkranz (1993), respectively. Water vapour absorption is calculated using the line parameters from the Liebe (1989) millimetre-wave propagation model (MPM89), with modifications to the line widths at 20 and 183GHz as proposed by Payne et al. (2008). The water vapour continuum absorption is calculated according to the laboratory measurements of Koshelev et al. (2011), where we refitted the parametrization to be consistent with the modified MPM89 absorption line parameters rather than the Rosenkranz (1998) parameters used in the Koshelev et al. study.

Within ARTS the bulk scattering properties of hydrometeors are calculated from the single scattering properties of particle entities (in ARTS denoted as scattering elements) and their respective vertical profiles of number concentration. Here, the scattering elements represent single ice crystals with a given maximum dimension $D_{\mathrm{max}}$ taken from the ARTS scattering database (Eriksson et al., 2018). This database provides single scattering properties for a wide range of ice crystal habits and sizes as described below. We also make use of the capability within ARTS to internally derive number concentrations for each scattering element using a parametrized particle size distribution (PSD), which defines the particle number density per unit size, and the sizes of the individual scattering elements. ARTS offers a number of PSD parametrizations, and here we use the Field et al. (2007) (F07) midlatitude PSD which we will show in sec. 5.1 is consistent with the in-situ observations. The F07

PSD is parametrized in terms of temperature and its second moment

$$M_2 = \int D_{\max}^2 N(D_{\max}) \mathrm{d}D_{\max}, \tag{1}$$

where $N(D_{\max})$ is the PSD. Typically, $M_2$ is related to the bulk ice water content (IWC) using a particle mass-dimension relationship, allowing vertical profiles of the PSD to be calculated from profiles of temperature and IWC. However, in this study the vertical profiles of $M_2$ are derived directly from lidar observations of particle extinction coefficient using an empirical relationship determined from the in-situ measurements as described in sec. 5.2. This approach is used because the various particle habits within the ARTS database have different mass-dimension relationships, so using an IWC-based approach would result in inconsistent size distributions when comparing simulations with different ice crystal habits. Directly using $M_2$ as input to ARTS ensures that the PSD is the same for each habit considered; however, it means that the simulations for different particle habits will correspond to different total ice masses. To ensure that, as far as possible, the clouds input to the radiative transfer simulations are consistent with the in-situ observations both in terms of particle size distribution and total ice mass, in-situ ice water content measurements from the Nevzorov probe are used to restrict the simulations to particle habits with realistic mass-dimension relationships as described in 5.1.

The ARTS scattering database provides single scattering properties for a wide variety of representative hydrometeor particle habits based on Discrete Dipole Approximation (DDA) calculations. The habits include both single pristine crystals and a number of different aggregates. For this study, we have selected particles from the database that are representative of cloud ice and snow (see Table 2). All particles are assumed to have random orientation, and simulations are performed independently for each particle habit. In reality clouds will consist of mixtures of different particle habits that will vary both horizontally and vertically throughout the cloud. However, we do not have sufficient information available from the observations to provide constraints on the spatial variation of habit mixtures, and we expect the brightness temperatures for habit mixtures to be contained within the range of values given by the individual habits. The particle habits are briefly summarised below; for full details refer to Eriksson et al. (2018).

The selected single crystal habits consist of ColumnType1 and PlateType1, which are hexagonal columns and plates respectively, with an aspect ratio that varies with particle size. The various bullet rosettes consist of hexagonal bullets connected at the tips and positioned perpendicular to each other. For the 3- and 4-armed rosettes, cases with all bullets arranged in a single plane (termed flat) or in multiple planes (termed perpendicular) are covered.

Six aggregate habits were generated through stochastic simulations using either hexagonal columns, blocks or plates as constituent crystals. Aggregates were created using two different sizes of constituent crystal, and are referred to as small (constituent crystal $D_{\max} = 100\mu m$ on average) and large (constituent crystal $D_{\max} = 350\mu m$ on average) aggregates, accordingly. Details of the aggregate generation process and micromorphology are available in Eriksson et al. (2018). Since the aggregates do not cover sizes smaller than the constituent crystals, they are combined with a corresponding single hexagonal crystal habit (column, block or plate) in a habit mixture. The transition between the single crystal and the aggregate is performed smoothly across a range of between 4 and 6 sizes by combining the two habits with a weighting that varies linearly with volume equiv-

alent diameter. The same mixing procedure has been applied for standard habits shipped with the ARTS database, and is also applied for all habits selected here that lack small size particles.

The SectorSnowflake was defined by Liu (2008) and is composed of three ellipsoids centred at a common point and angled at 60°, forming a snow-like particle. The EvansSnowAggregate by Evans et al. (2012) was generated in a similar fashion to the ARTS database aggregates, but uses dendritic snow crystals as building blocks. Finally, the IconSnow habit is intended to mimic snow, and follows the snow mass-dimension parametrization of the ICON (Icosahedral non-hydrostatic general circulation model), which is a part of the German HD(CP)$^2$ project (Zängl et al., 2015). It is generated by applying dipoles successively in cylindrical layers. This habit is used as habit mixture, combined with hexagonal columns at lower sizes. For this study, we removed two sizes from the database data in order to ensure unique sizes in $D_{\mathrm{max}}$ as well as a monotonic relationship between $D_{\mathrm{max}}$ and the volume equivalent diameter $D_{\mathrm{veq}}$; the resulting modified habit is denoted as IconSnow*.

An additional particle habit called the 6HexagonalRosette is not part of the standard ARTS database, but has been created specifically for this study to try and match the particles observed during B895. It is composed of three equal hexagonal columns arranged perpendicular to each other and sharing a common centre. The dimensions of the columns were determined numerically to match the Brown and Francis (1995) mass-dimension relationship as closely as possible. However, this is not possible for sizes below $\sim$170µm unless super-density is allowed. This habit is therefore used as a habit mixture with block columns at lower sizes.

For flight B895, the sea surface properties are represented using the FASTEM model (Liu et al., 2011), with surface temperature and windspeed taken from the UKV NWP model. Note that FASTEM is only valid for frequencies up to 183GHz, but at higher frequencies the sensitivity to the surface will be small due to increased water vapour absorption and emission. However, at 243GHz and 325±9.5GHz there is likely to be some remaining surface sensitivity, particularly for dry atmospheres. For the land surfaces in flight B939 a fixed emissivity of 0.95 is used, with the surface temperature taken from the UKV NWP model.

Simulations were performed for idealised nadir-pointing pencil beams at the nominal centre frequencies of the dual pass-bands for each MARSS/ISMAR channel. No antenna pattern or other sensor response characteristics were included. Tests using clear-sky simulations with representative sensor responses suggest that the error due to the neglect of the full sensor response is small, of order 0.1K. For the nadir views used in this study the error due to the neglect of the finite antenna beamwidth is also much less than 0.1K, but the impact of finite beamwidth would be much greater if off-nadir views were used.

## 5 Cloud microphysical properties

As discussed in the previous section, a number of assumptions are required to determine the vertical profiles of the cloud scattering and absorption properties representative of the clouds at the time and location of the MARSS and ISMAR observations. In particular, a parametrization for the PSD is required, along with a relationship between the lidar extinction coefficient and the control variable for the PSD parametrization, here chosen (for consistency with F07) as the value of the second moment $M_2$. Appropriate choices of particle habit are also required. Our approach in this study is to ensure that these assumptions are all consistent with in-situ observations from the same cloud, despite being obtained at different times than the MARSS/ISMAR

observations. Specifically, we will show that the F07 midlatitude parametrization provides a reasonable representation of the in-situ PSDs and identify particle habits, which have a mass-dimension relationship that gives consistency between ice water content values derived from the PSDs and bulk measurements from the Nevzorov probe. We also use the in-situ PSDs to derive an empirical relationship between the lidar extinction coefficient $\sigma$ and the second moment of the PSD, $M_2$.

## 5.1 Particle size distributions and habits

The mean particle size distributions measured during each in-situ run are shown for the two flights in figs. 3 and 4, along with example particle images. Also plotted are the tropical and midlatitude PSD parametrizations from F07 using the run-mean temperature and the same values of $M_2$ as the observed run-mean PSDs. For comparison the McFarquhar and Heymsfield (1997) (MH97) parametrization is also shown. This parametrization is expressed in terms of the ice water content and the particle mass-equivalent dimension. For these plots the MH97 PSD was calculated using the run-mean IWC from the Nevzorov probe, and $D_{\mathrm{max}}$ was related to the mass-equivalent dimension using the Brown and Francis (1995) mass-dimension relationship, which in SI units is given by

$$m = 0.019 D_{\mathrm{max}}^{1.9}. \tag{2}$$

At cold temperatures close to the cloud tops the PSDs are typically dominated by small particles. The PSDs become broader at higher temperatures with increasing concentrations of larger particles and the development of a bimodal structure with a distinct peak around 200µm. However, as discussed by O'Shea et al. (2016) the small-particle mode may be an instrument artefact. The MH97 parametrization significantly over-estimates the concentration of small particles with sizes less than 100µm, and frequently under-estimates the concentration of particles at larger sizes, which are responsible for the majority of the cloud ice mass. The tropical version of the F07 parametrization also tends to over-estimate the concentration of small particles, particularly at warmer temperatures. The best match to the observations up to around 500µm is given by the midlatitude F07 parametrization. However, at the warmer temperatures and largest sizes it tends to fall off too rapidly, and the broader tropical parametrization gives a better match for particle sizes of order 1000µm. For the cases studied here, these large particles generally only contribute a small proportion of the ice mass, so the midlatitude F07 parametrization has been selected as the most representative model for the radiative transfer simulations.

Particle habits are important for the radiative transfer simulations as differences in single scattering properties can significantly affect the bulk scattering properties for a given size distribution (Eriksson et al., 2015; Evans and Stephens, 1995b). The particle images for flight B895 show a variety of habits including pristine columns and rosettes, quasi-spherical particles and aggregates. O'Shea et al. (2016) applied an automatic habit recognition algorithm to the CPI images from this flight and found that at sizes less than 100µm the dominant habit was quasi-spherical particles, but for sizes greater than 100µm the majority of particles consisted of aggregates and rosettes. The interpretation of B939 is more difficult as only low resolution CIP-15 images, which are less suitable for habit classification, are available. However, columns, rosettes and aggregates can all be seen in the images in fig. 4 at 226K, whilst at the warmer temperatures aggregates are more dominant.

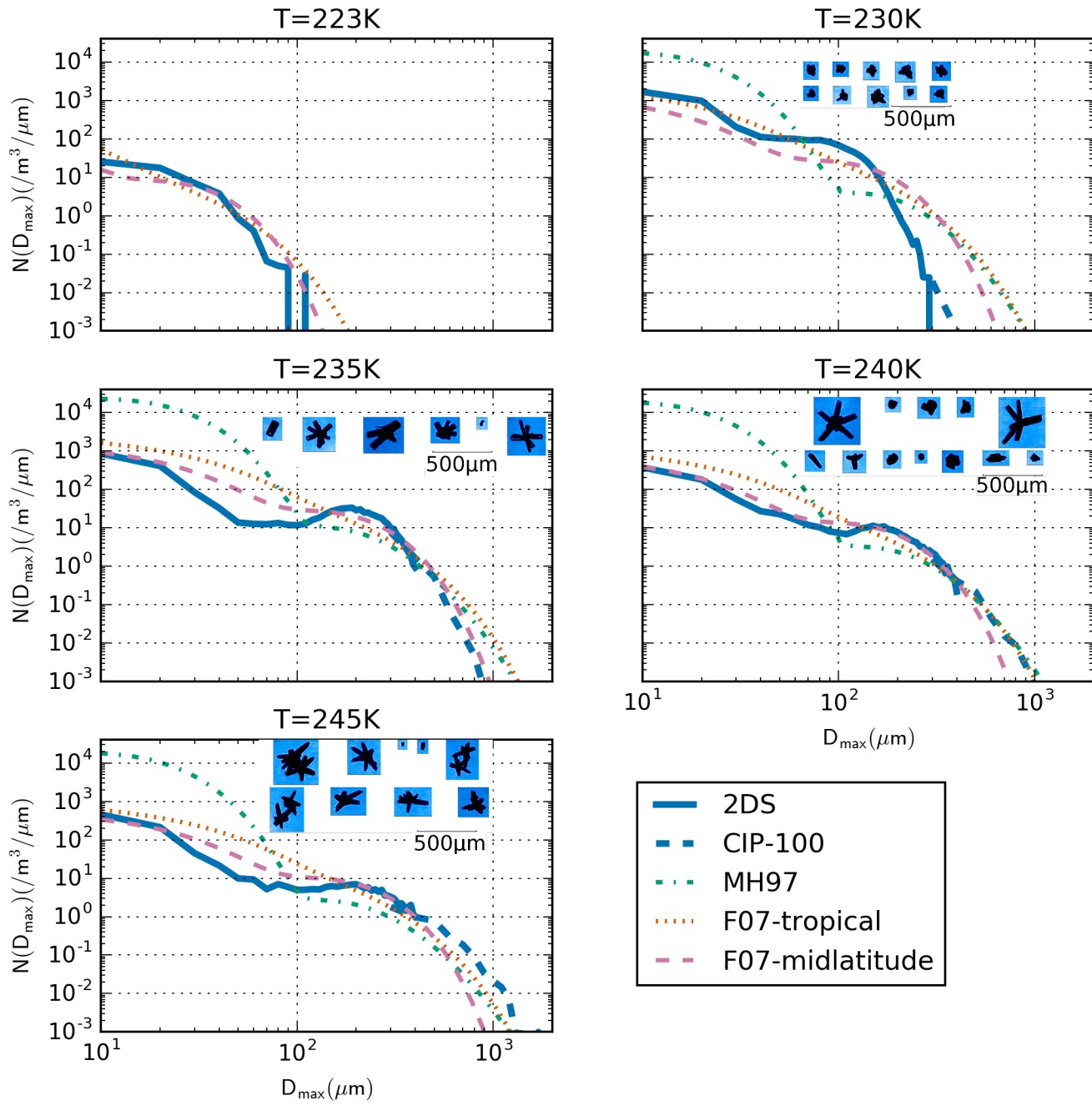

**Figure 3.** Particle size distributions from in-situ runs during flight B895, with example CPI images showing ice crystal habits.

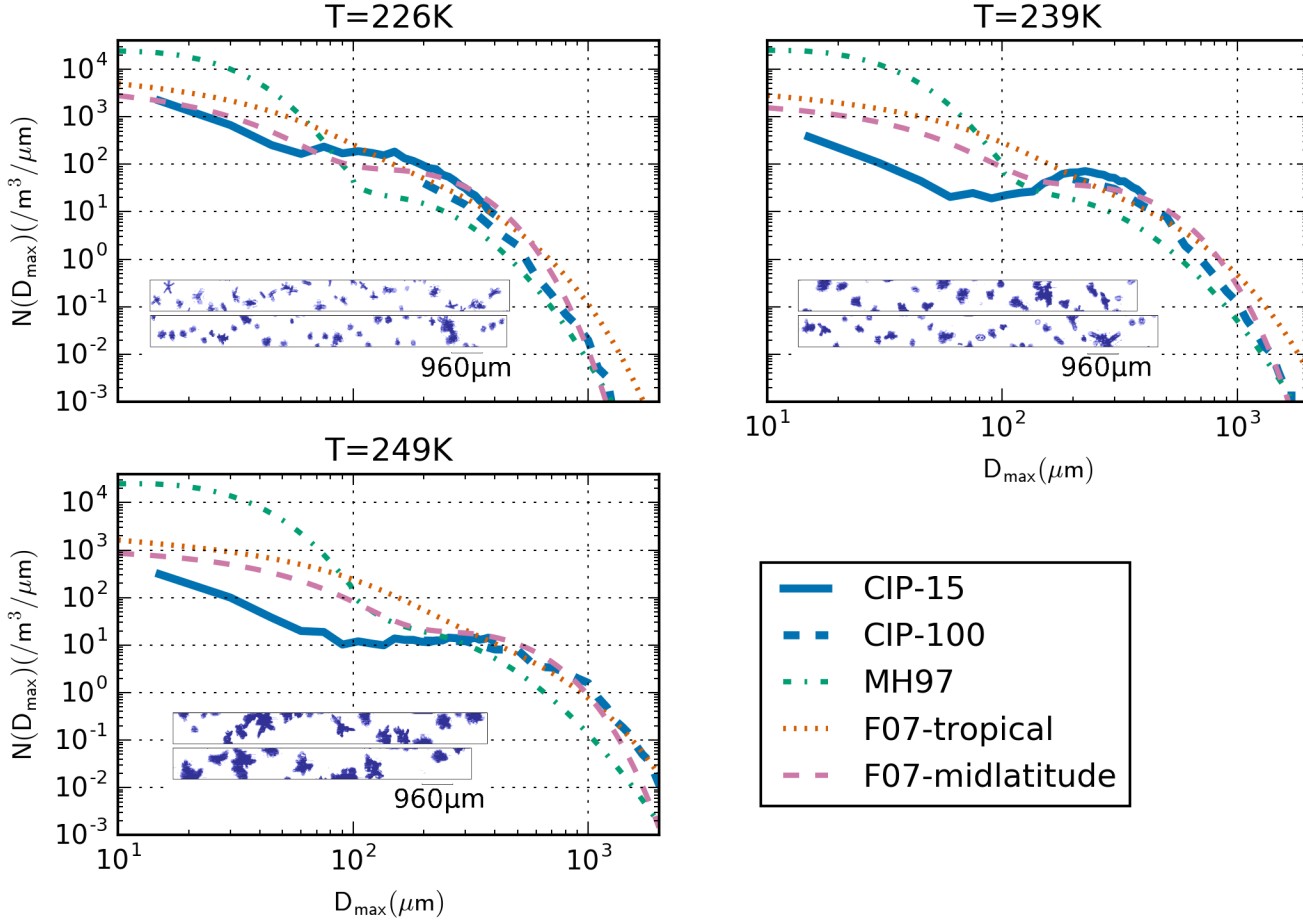

**Figure 4.** Particle size distributions from in-situ runs during flight B939, with example CIP-15 images showing ice crystal habits.

For simplicity, in this study the radiative transfer simulations are performed using single particle habits from the ARTS database. As discussed in the previous section no single habit can completely model the variety of ice crystals observed in real clouds which generally contain mixtures of particle habits that vary both horizontally and vertically. Nevertheless, it is expected that the optical properties of a single habit, when integrated over a range of sizes, can be sufficiently representative to permit realistic brightness temperature simulations. This approach was adopted at lower frequencies by Geer and Baordo (2014), who demonstrated that a single snowflake habit could successfully reproduce observed brightness temperatures at a global scale between 10 and 183GHz.

The in-situ observations can be used to apply a constraint on the habits selected for the simulations based on the bulk ice water content. Each of the particle habits in the database has an associated mass-dimension relationship $m(D_{max})$ which, when

| Particle Habit | B895 | B939 |
|---|---|---|
| ColumnType1 | **0.5** | **1.5** |
| PlateType1 | 0.3 | **1.2** |
| 6-BulletRosette | 0.3 | **1.1** |
| 5bulletRosette | 0.3 | **0.9** |
| Perpendicular4-BulletRosette | 0.2 | **0.7** |
| Flat4BulletRosette | 0.2 | **0.7** |
| Perpendicular3BulletRosette | 0.2 | **0.7** |
| Flat3-BulletRosette | 0.2 | **0.6** |
| 6HexagonalRosette | **0.9** | 2.4 |
| SectorSnowflake | **1.6** | **3.8** |
| SmallBlockAggregate | 0.3 | **1.0** |
| SmallColumnAggregate | 0.1 | 0.3 |
| SmallPlateAggregate | 0.2 | **0.7** |
| LargeBlockAggregate | **1.2** | 3.4 |
| LargeColumnAggregate | 0.2 | **0.6** |
| LargePlateAggregate | **0.6** | 1.9 |
| EvansSnowAggregate | 0.4 | **0.8** |
| IconSnow* | **0.9** | 2.6 |

**Table 2.** Mean IWC ratio for each particle habit and flight. The habits selected for the simulations are highlighted in bold.

combined with the observed particle size distribution, will determine the ice water content according to

$$IWC = \int m(D_{\max})N(D_{\max})\mathrm{d}D_{\max}. \tag{3}$$

For consistency, we should ensure that the ice water content derived according to eq. (3) for the selected model particles matches the bulk measurement from the Nevzorov probe. This will ensure that the simulations remain consistent with the observations both in terms of particle sizes and total mass. The ratio of the two IWC values is plotted in fig. 5 for the two flights, as well as the IWC calculated from eq. (3) using the Brown and Francis (1995) mass-dimension relationship eq. (2). To make the plot clearer the data have been binned into ranges according to the Nevzorov IWC and the mean value in each bin is shown. Table 2 presents the mean ratio of the two IWC values for all PSDs where the Nevzorov IWC is greater than $0.005\mathrm{gm}^{-3}$.

For B895, the observations are consistent with the Brown and Francis relationship, but this is not the case for B939, where particles with smaller mass for a given size are required to give consistency between the two IWC values. For the simulations we select the particles where the mean IWC ratio lies between 0.5 and 1.5, i.e. on average the IWC predicted by integrating the individual particle masses over the observed PSD lies within $\pm50\%$ of the bulk IWC measured by the Nevzorov probe.

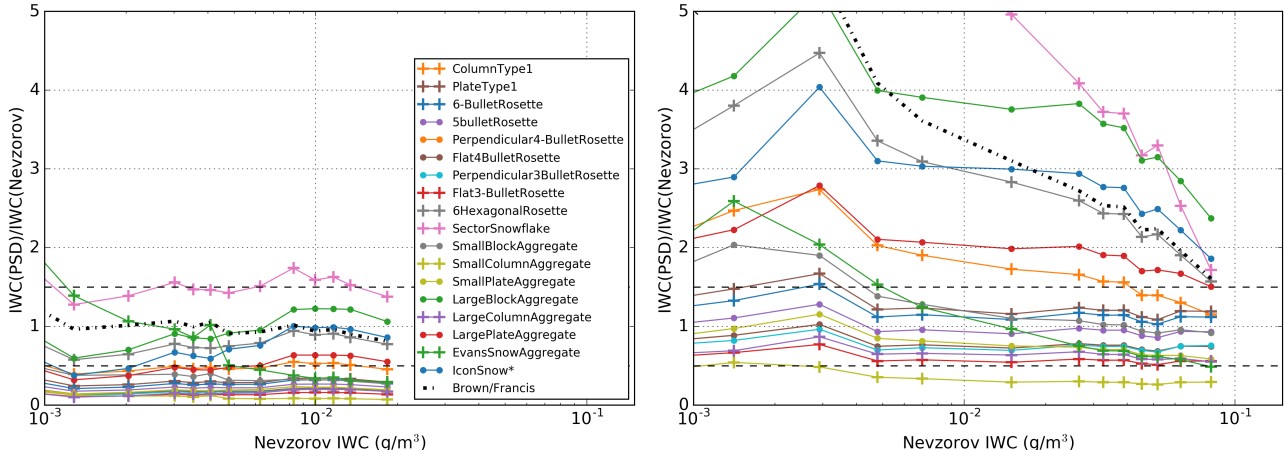

**Figure 5.** Ratio of IWC derived from the in-situ PSDs for each particle habit with bulk IWC measured using the Nevzorov probe for B895 (left) and B939 (right). The dash-dot line uses the Brown and Francis m-D relationship and the dashed horizontal lines indicate $\pm 50\%$.

For both flights, we also perform simulations with the SectorSnowflake as this was the habit which Geer and Baordo (2014) found to give the best results at frequencies up to 183GHz, although it should be noted that its mass is rather inconsistent with the Nevzorov IWC for B939. For B895, the selected habits are the ColumnType1, 6HexagonalRosette, LargeBlockAggregate, LargePlateAggregate and IconSnow. The best IWC agreement is given by the 6HexagonalRosette, which has been specifically
constructed to closely follow the Brown and Francis mass-dimension relationship. For B939, the selected habits are the ColumnType1, PlateType1, SmallBlockAggregate, SmallPlateAggregate, LargeColumnAggregate, EvansSnowAggregate, and all the bullet rosettes. However, since all the bullet rosettes have similar scattering properties, for clarity we will only show results for the 6-BulletRosette and the Flat3-BulletRosette which give the upper and lower bounds of simulated brightness temperature for all the bullet rosette habits, as well as giving the upper and lower bounds of IWC.

**5.2   Relationships between lidar extinction, $M_2$ and IWC**

The in-situ measurements have also been used to derive relationships between the volume extinction coefficient at 355nm and the second moment of the PSD, $M_2$, for the two flights. These relationships are used to retrieve profiles of $M_2$ from the lidar observations as described above. For ice particles which are much larger than the lidar wavelength the volume extinction coefficient $\sigma$ can be estimated using the geometric optics approximation:

$$\sigma = 2 \int \langle A(D_{\max}) \rangle N(D_{\max}) \mathrm{d}D_{\max}, \tag{4}$$

where $\langle A(D_{\max}) \rangle$ is the mean projected particle area as a function of size and $N(D_{\max})$ is the particle number density. For flight B895 the high spatial resolution images from the CPI were used to determine a power-law relationship between projected area and particle size, which is given in SI units by $\langle A(D_{\max}) \rangle = 0.025 D_{\max}^{1.664}$. This was used in eq. 4 with the 2DS PSDs averaged over 10s intervals to derive the volume extinction coefficient $\sigma$ associated with each PSD. Tests using the CIP-100

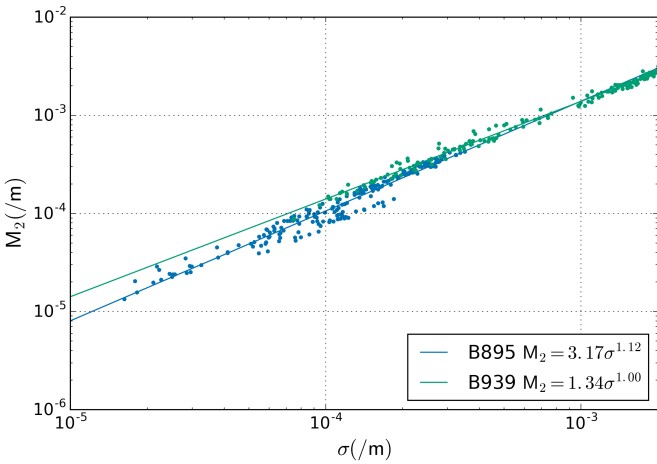

**Figure 6.** Extinction/$M_2$ relations determined from in-situ measurements.

PSDs showed that particles larger than the size range measured by the 2DS made a negligible contribution to both $\sigma$ and $M_2$ for this cloud. CPI and 2DS images are not available for flight B939, so instead $\sigma$ and $M_2$ were determined directly from the lower resolution CIP-15 and CIP-100 measurements. The CIP-15 was used for sizes up to 650μm and the CIP-100 was used for the larger sizes. The size threshold between the two instruments was selected to ensure that the CIP-100 particles occupied

a sufficient number of pixels to allow the projected area to be determined.

Figure 6 shows the relationship between the extinction and the second moment of the PSD for the two flights. Also shown are power-law fits between the two quantities which are used to determine the vertical profile of $M_2$ from the lidar observations. Although we have used separate fits for each flight the results are similar, and a single fit could be applied to both flights. Note that significantly larger values of $\sigma$ and $M_2$ were encountered in B939 compared to B895.

A similar technique was also used to derive empirical relationships between the volume extinction coefficient and the ice water content for each flight by fitting a power-law to the Nevzorov bulk IWC as a function of $\sigma$ derived using geometric optics and the data from the optical array probes as described above. This allows the ice water path to be retrieved from the lidar profiles. The resulting fits for the two flights were $IWC = 27.17\sigma^{0.94}$ for B895 and $IWC = 90.06\sigma^{1.21}$ for B939. The B939 result is consistent with the results of Heymsfield et al. (2005), whilst the B895 result produces a significantly greater IWC for

a given value of $\sigma$ due to a difference in the cloud microphysical properties. The resulting distributions of IWC derived from the lidar data are shown in fig. 7. Although these IWC profiles are not directly required for the radiative transfer simulations, they are used to determine the ice water path which is used in the following section.

## 6   Results and discussion

In this section we compare the observed brightness temperatures with the radiative transfer simulations. Results are shown

for the particle habits which give an ice mass that is consistent with the in-situ observations as described above. Figures

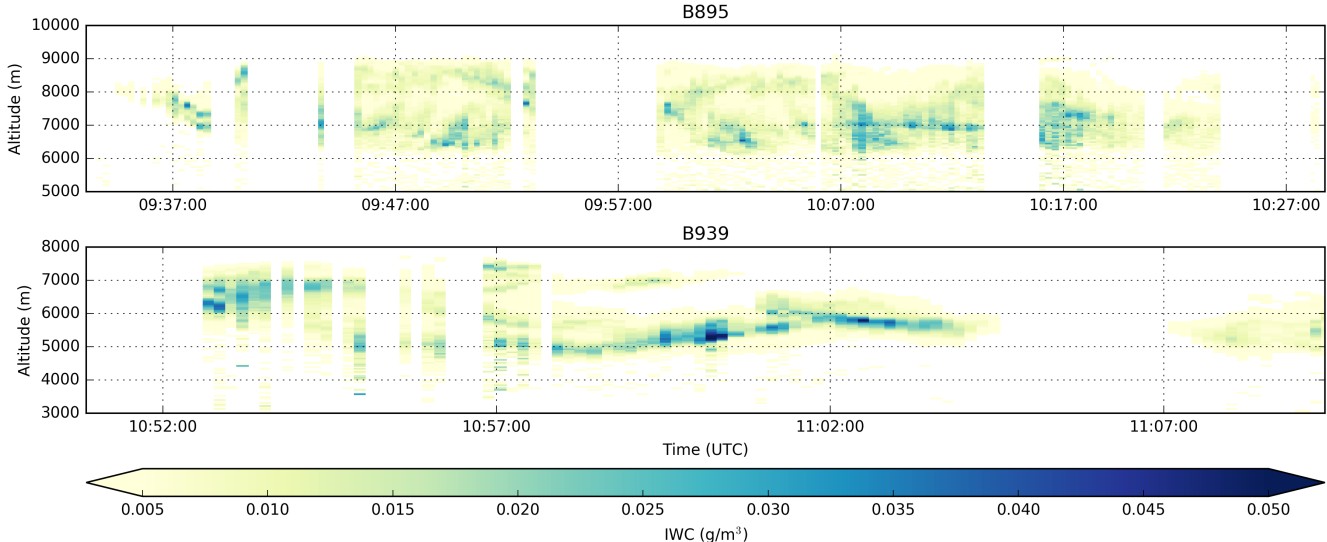

**Figure 7.** Lidar-derived ice water content profiles. Missing data correspond to aircraft turns or times when the lidar did not penetrate the full cloud depth.

8 and 9 show, for each of the two flights, the difference between the observed or simulated brightness temperatures and a clear-sky simulation using the same temperature and water vapour profiles as the cloudy simulations. To simplify the plots we focus on the higher frequency channels where there is greatest sensitivity to cloud ice. Only the H-polarised channel is shown for 664GHz as it has lower noise and both polarisations are expected to be equivalent for nadir views. We also ignore

the 448±1.4GHz channel which has a rather high-peaking clear-sky weighting function and so shows little sensitivity to all but the highest clouds, and the 243GHz channels because they have significant surface contributions. The error bars shown for the observations are a combination of both the estimated radiometer noise and maximum bias calculated as described in Fox et al. (2017), although for some channels such as 448±7.2GHz the scatter of the observations suggests that this approach underestimates the uncertainty. The mean and RMS differences between the observed and simulated brightness temperatures

for each particle habit are shown in tab. 3. Note that the SectorSnowflake habit and, for B895, the PlateType1 habit have been included in the figures and table for comparison purposes as described below in spite of the fact that they do not give a mass consistent with the in-situ observations.

     As expected the observations show that the clouds cause brightness temperature depressions compared to the clear-sky case, and the simulations follow the main trends seen in the observations. The brightness temperature depressions increase with

frequency due to stronger scattering. They are also greater for channels further away from absorption line centres because of the warmer background temperature and sensitivity to a greater depth of cloud. The observed brightness temperatures are generally within the range of values simulated for the different ice crystal habits, with the notable exception of the 664GHz observations for flight B895, where the observed brightness temperatures depressions are considerably smaller than all of the

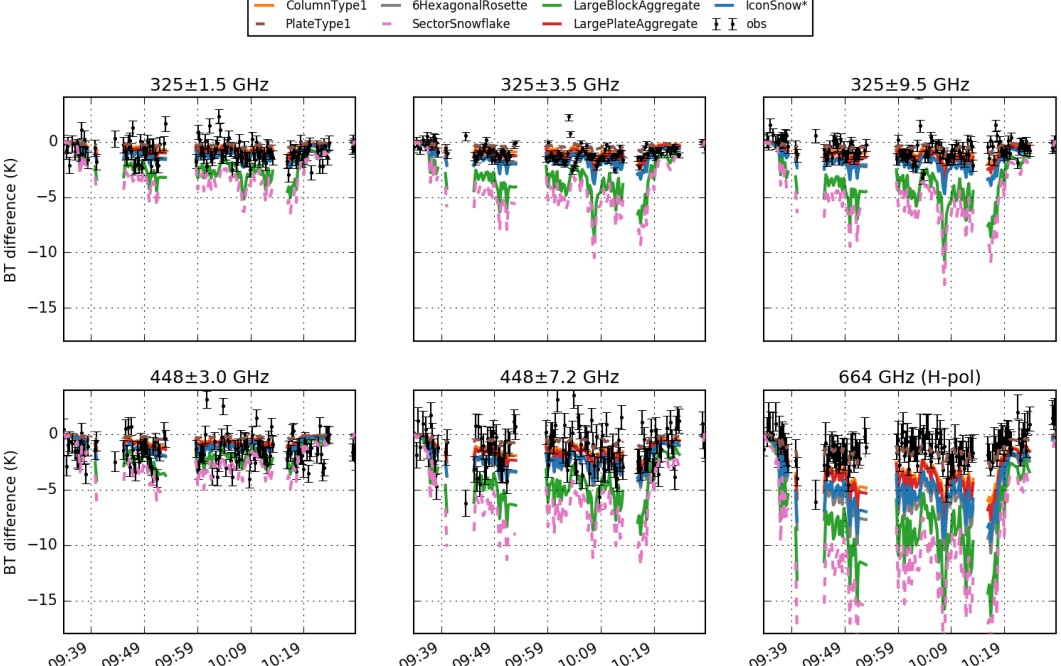

**Figure 8.** Time-series of measured and simulated cloud-induced brightness temperature differences for B895. The dashed lines indicate habits which are not consistent with the in-situ bulk mass.

simulations for habits with masses consistent with the in-situ observations. The particle habit can have a considerable effect on the brightness temperature depression, with extreme differences of over 10K at 664GHz between the simulations for different habits with similar bulk masses. These differences are primarily caused by differences in the particle single scattering properties as we have ensured that both the size distribution and the bulk ice mass are approximately the same for the majority of the particle habits shown.

The observed and simulated brightness temperature depressions are also plotted in figs. 10 and 11 as a function of the cloud ice water path. The ice water path was calculated from the lidar extinction profiles using the empirical relationships between the extinction and the ice water content described in sec. 5.2. The observed cloud signals for B895 are generally rather small, even at 664GHz for the largest values of IWP where, with the exception of a few outliers, they are around 2-3K. The simulations are not inconsistent with the observations at 325 and 448GHz, although the brightness temperature depressions at these frequencies are small compared to the observation uncertainties. However, at 664GHz none of the simulations for the selected habits are consistent with the observations, with the model predicting significantly larger brightness temperature depressions than were observed, particularly at the largest values of IWP. For comparison we have also included simulations using the PlateType1 habit which gives a much closer match to the observations at 664GHz for this flight. This demonstrates that the observed brightness temperature depressions are within the range of values that might be expected given realistic particle sizes and scattering properties. However, the PlateType1 habit is not consistent with the in-situ observations of bulk

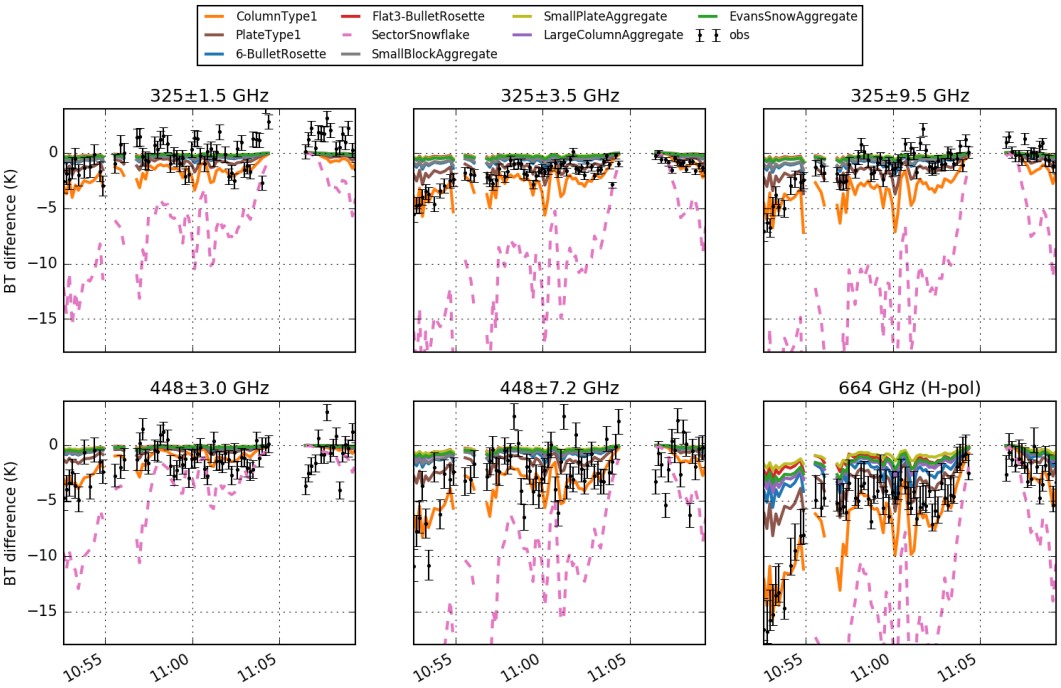

**Figure 9.** Time-series of measured and simulated cloud-induced brightness temperature differences for B939. The dashed lines indicate habits which are not consistent with the in-situ bulk mass.

ice mass, as it results in an IWC that is only one third of the observed IWC. We have therefore been unable to obtain fully consistent simulations for the 664GHz channel for flight B895. One possible explanation for this inconsistency could be the presence of oriented particles. The presence of horizontally aligned particles would affect many parts of the analysis presented here, including the particle bulk scattering properties and hence the simulated brightness temperatures. However, they could

also have a critical impact on the lidar volume extinction coefficient, and might be expected to cause increased extinction for a given PSD or ice mass. We could therefore significantly overestimate the particle sizes (and hence ice mass) in the input to the radiative transfer model in the presence of oriented particles. Dual-polarised measurements for off-nadir views could be used to determine the presence of oriented particles, but calibration problems and excess noise on the vertically polarised 664GHz receiver precludes this analysis and the full consideration of the effects of particle orientation is beyond the scope of this study.

In future, the addition of dual-polarised receivers at 874GHz to ISMAR will allow further study of the impact of oriented particles. Another possibility for the discrepancy between the observations and simulations at 664GHz is that there was a significant change in the cloud microphysical properties between the time of the MARSS/ISMAR observations and the in-situ runs for this flight. However, there is no obvious way to confirm if this is the case using the instruments currently available on the aircraft. Combining lidar and cloud radar observations could provide additional constraints on the cloud microphysics

co-incident in time with the MARSS/ISMAR observations.

| Flight | Habit | 325±1.5GHz | | 325±3.5GHz | | 325±9.5GHz | | 448±3.0GHz | | 448±7.2GHz | | 664GHz (H-pol) | |
|---|---|---|---|---|---|---|---|---|---|---|---|---|---|
| | | Mean | RMS | Mean | RMS | Mean | RMS | Mean | RMS | Mean | RMS | Mean | RMS |
| B895 | ColumnType1 | -0.3 | 1.0 | -0.1 | 0.7 | 0.3 | 1.0 | -0.9 | 1.6 | 0.1 | 1.7 | 1.7 | 2.2 |
| | *PlateType1* | *-0.6* | *1.1* | *-0.5* | *0.9* | *-0.2* | *0.9* | *-1.2* | *1.8* | *-0.7* | *1.8* | *-0.2* | *1.1* |
| | 6HexagonalRosette | 0.2 | 1.0 | 0.6 | 1.0 | 1.1 | 1.6 | -0.4 | 1.4 | 1.0 | 2.1 | 3.7 | 4.2 |
| | *SectorSnowflake* | *2.2* | *2.7* | *3.4* | *3.9* | *4.4* | *5.1* | *1.2* | *2.1* | *4.6* | *5.5* | *8.5* | *9.5* |
| | LargeBlockAggregate | 1.4 | 2.0 | 2.3 | 2.9 | 3.2 | 3.8 | 0.4 | 1.6 | 3.0 | 3.8 | 6.2 | 7.0 |
| | LargePlateAggregate | -0.1 | 1.0 | 0.1 | 0.8 | 0.6 | 1.2 | -0.8 | 1.5 | 0.3 | 1.8 | 2.0 | 2.5 |
| | IconSnow* | 0.2 | 1.0 | 0.6 | 1.1 | 1.2 | 1.7 | -0.5 | 1.4 | 0.9 | 2.0 | 3.1 | 3.6 |
| B939 | ColumnType1 | 1.7 | 2.3 | 0.7 | 1.3 | 1.8 | 2.4 | -0.2 | 1.4 | 0.7 | 2.5 | 0.8 | 2.3 |
| | PlateType1 | 0.9 | 1.8 | -0.5 | 1.0 | 0.3 | 1.5 | -0.8 | 1.6 | -1.0 | 2.6 | -2.2 | 3.3 |
| | 6-BulletRosette | 0.6 | 1.7 | -1.1 | 1.5 | -0.4 | 1.6 | -1.0 | 1.7 | -1.6 | 2.9 | -3.2 | 4.3 |
| | Flat3-BulletRosette | 0.3 | 1.7 | -1.5 | 1.8 | -0.8 | 1.9 | -1.2 | 1.8 | -2.0 | 3.2 | -4.1 | 5.2 |
| | *SectorSnowflake* | *6.4* | *7.2* | *7.8* | *8.9* | *10.4* | *11.6* | *2.5* | *3.8* | *7.6* | *9.4* | *10.4* | *12.3* |
| | SmallBlockAggregate | 0.5 | 1.7 | -1.2 | 1.5 | -0.5 | 1.7 | -1.0 | 1.7 | -1.7 | 3.0 | -3.7 | 4.8 |
| | SmallPlateAggregate | 0.3 | 1.7 | -1.5 | 1.8 | -0.8 | 1.9 | -1.2 | 1.8 | -2.1 | 3.3 | -4.4 | 5.5 |
| | LargeColumnAggregate | 0.4 | 1.7 | -1.4 | 1.7 | -0.7 | 1.8 | -1.1 | 1.8 | -1.9 | 3.1 | -3.7 | 4.7 |
| | EvansSnowAggregate | 0.4 | 1.7 | -1.4 | 1.7 | -0.8 | 1.8 | -1.1 | 1.8 | -1.9 | 3.2 | -3.9 | 5.0 |

**Table 3.** Mean and RMS difference between observed and simulated brightness temperatures for each ice crystal habit. The habits in italics are not consistent with the in-situ bulk mass but have been included for comparison purposes.

It is also of interest to consider the cloud signal simulated for the SectorSnowflake habit, which was used successfully to reduce first-guess departures at a global scale in the ECMWF forecast model for frequencies up to 183GHz by Geer and Baordo (2014). For flight B895 it results in brightness temperature depressions that are significantly larger than the observations for all the frequencies considered, despite the fact that it predicts an IWC that is only 60% greater than the in-situ bulk measurements. The simulated cloud signals are the most extreme of all the habits considered for this flight, with mean biases ranging from 1.2 to 8.5K depending on frequency. This indicates that the SectorSnowflake habit has significantly too much scattering for a given crystal mass and size for this particular cloud.

The cloud-induced brightness temperature depressions during B939 are significantly larger than for B895, particularly at the start of the run which corresponds to both the largest IWP and highest altitude cloud. There is a large spread in simulated cloud signals for the different particle habits, with the ColumnType1 giving the largest cloud signal and the SmallPlateAggregate giving the smallest cloud signal. In the early part of the run the observations at all frequencies are generally consistent with the ColumnType1 habit. However, beyond 10:55 there is no single habit which is consistent with the observations across all frequencies. In particular the 325±3.5 and 664GHz observations generally lie between the ColumnType1 and PlateType1 habits whilst the 325±9.5GHz observations are more consistent with the aggregate and rosette habits. It is difficult to draw

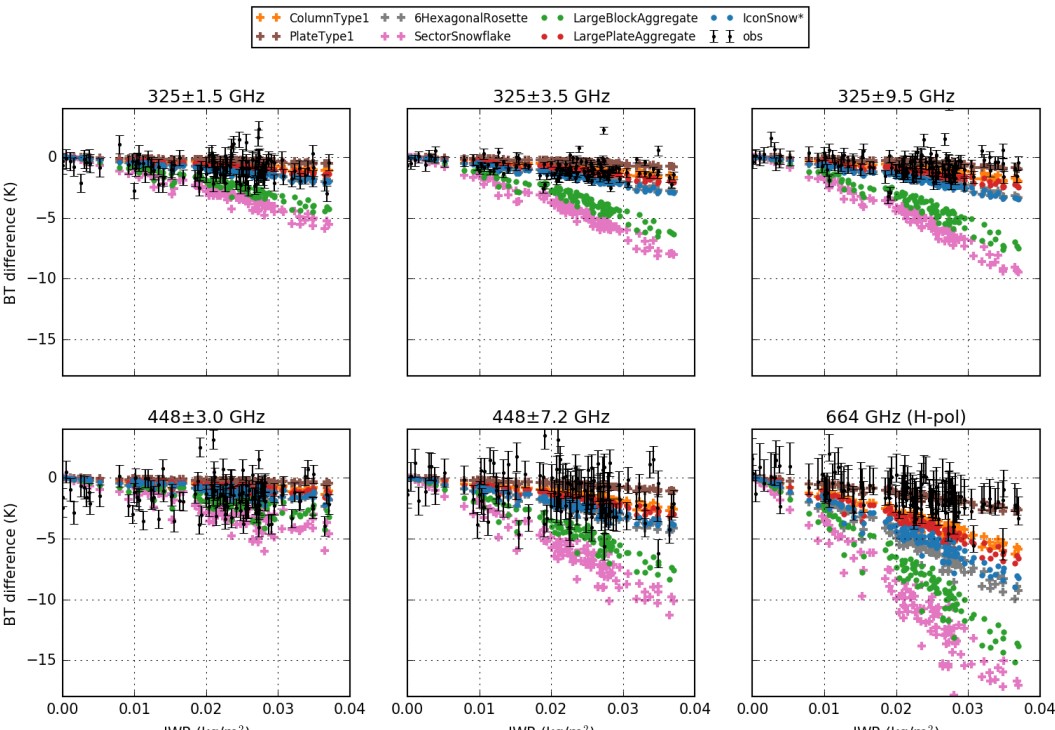

**Figure 10.** Cloud-induced brightness temperature differences as a function of ice water path for B895.

firm conclusions from the 325±1.5 observations as fig. 11 shows a warm bias of around 2K in the observations compared to the simulations at low ice water paths which may be due to errors in the water vapour profile or instrument calibration errors. The large scatter in the 448GHz observations also makes it difficult to identify clear trends. The difference in behaviour between the channels could be caused by a change in particle habit through the depth of the cloud, with more single ice

5   crystals at the cloud top and larger aggregates towards the cloud base as suggested by the particle images in fig. 4. It may be possible to represent this by using variable mixtures of different habits, similar to approach adopted by Baran et al. (2014), who used a temperature-dependent habit mixture to represent ice crystal scattering properties. Since the different channels will be most sensitive to cloud ice at different altitudes this approach could improve the consistency between the observations and simulations across the frequency range. The cloud signal from a habit mixture will lie between the extremes of the single

10   habits forming the mixture, so it seems likely that a layered model consisting of a mixture of plates and columns at cold temperatures and aggregates at warmer temperatures would be capable of reproducing the observed brightness temperatures for B939. However, given the spatial and temporal differences between the remote-sensing and in-situ cloud measurements and the limited information on cloud microphysical properties available from the lidar extinction it is difficult to justify applying such a detailed model to these observations.

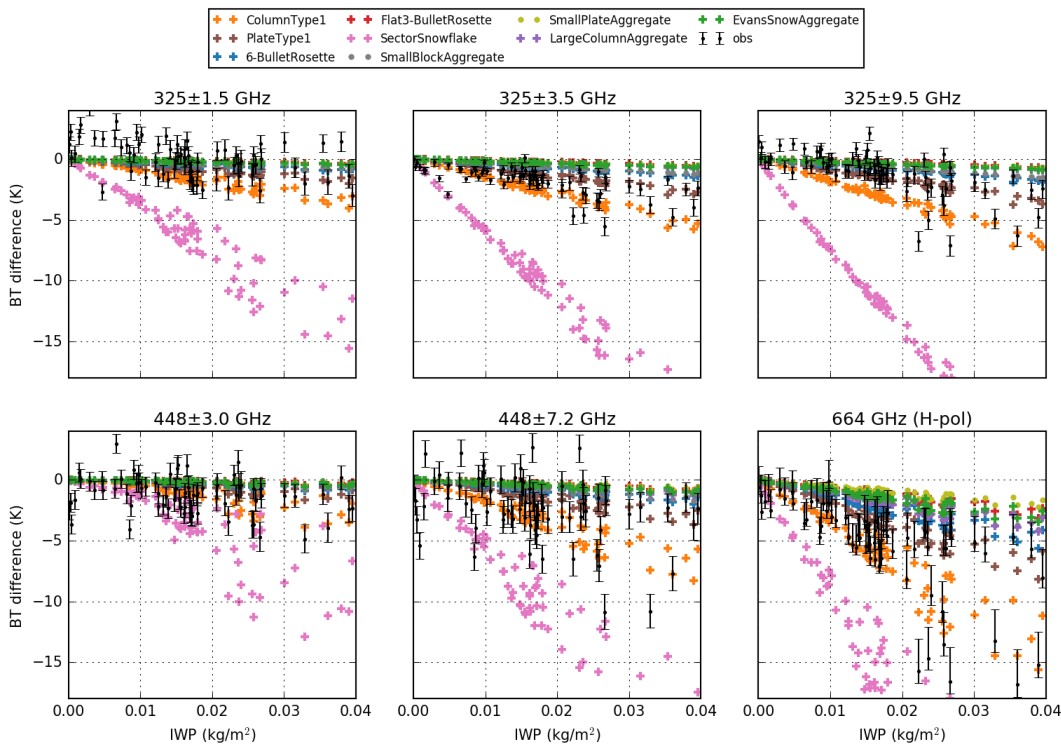

**Figure 11.** Cloud-induced brightness temperature differences as a function of ice water path for B939.

It is interesting to note that for B939 the observations in fig. 11 show a wide range of observed brightness temperature depressions for IWP greater than $0.02\mathrm{kg/m^2}$, and they are generally consistent with the simulations for either the ColumnType1 or the PlateType1 habits. The larger brightness temperature depressions which follow the ColumnType1 simulations correspond to times prior to 10:55 whilst the smaller brightness temperature depressions which are closer to the PlateType1 simulations

correspond to later times, including around 11:00. As shown by the lidar plots in fig. 2 this change in behaviour is associated with a distinct change in the cloud mean altitude and a lowering of the cloud top. However, the change in cloud signal cannot simply be attributed to the change in cloud height as the simulations for a given particle habit, which also account for the different cloud heights, do not show such a wide range of brightness temperature depressions. This suggests that the change in cloud signal is likely to be due to a difference in particle habit between the two cloud layers, although as we have only used

run-mean observations to select the most suitable PSD parametrization we cannot rule out spatial variations in the PSD.

Considering the SectorSnowflake habit for B939, it is clear that it greatly over-predicts the cloud scattering at all frequencies, and it predicts significantly greater brightness temperature depressions than all the other habits considered for this flight. In part this is due to the inconsistent mass for this habit; for the in-situ observations the bulk IWC obtained from eq. (3) using the SectorSnowflake particle mass was almost four times the value obtained from the Nevzorov probe. Since the PSD in the

radiative transfer simulations is fixed (through the vertical profiles of the second moment) this means that the IWP in the

simulations for the SectorSnowflake is also much larger than for the other habits. Note that our approach, which fixes the PSD and allows the ice mass to be determined according to the particle habit, differs from that of Geer and Baordo (2014), who define vertical profiles of ice water content and use this (along with the F07 tropical PSD parametrization) to determine the PSD according to the mass-dimension relationship of the habit of interest. Their approach will result in smaller particles for
this case, and hence reduced scattering signals which will be more consistent for the observations. However, it remains the case that the SectorSnowflake habit is unable to give consistency with both the observed ice mass and PSD for this flight.

It is instructive to compare the observed cloud signals between the two flights, which both cover similar ranges of ice water path. For example in B895 an IWP of $0.02\text{kg/m}^2$ led to a brightness temperature depression close to 1K at 664GHz whereas the same IWP gave a 6K brightness temperature depression for B939. This is likely to be caused by the differences in the
cloud microphysics for the two flights, although errors in the lidar IWP for B895 due to oriented particles cannot be ruled out. For B895 the cloud has a small ice water content but has a relatively large vertical extent, whereas for B939 it has a lower vertical extent but with higher IWC. This results in larger particles during B939 as shown by the PSDs in figs. 3 and 4, which cause more scattering for the same total ice mass. The cloud-induced brightness temperature depressions therefore appear to be sensitive to both the total ice mass (IWP) and the distribution of mass within the cloud. Whilst some information on the
vertical distribution of the ice mass will be contained in the observations at different frequencies there are implications for the minimum IWP that will be detectable by ICI, which will depend on the distribution of the ice mass throughout the atmospheric column, as well as the ice particle habits.

## 7   Conclusions

In this paper we have performed a closure study comparing millimetre and sub-millimetre passive observations to radiative
transfer simulations for two cirrus cloud cases. These two cases represent the best available FAAM aircraft observations for such a study, and care was taken to ensure that the inputs to the radiative transfer model were as realistic as possible and were consistent with in-situ measurements of the same clouds. In particular, we used lidar observations to constrain the vertical profiles of the particle size distributions, and ensured that both the parametrized PSD and the mass-dimension relationship of the selected ice crystal habits were consistent with in-situ observations.
We found that the measured brightness temperatures were often within the range of values simulated by the radiative transfer model using different particle habits, and the simulations were able to reproduce much of the variability in observed cloud signals. However, the cloud signals were mostly rather small for all but the highest frequency channels and IWP values, indicating that these cases represent the lower limit of ice cloud that can be detected using passive sub-millimetre observations given the measurement uncertainties. For B939 we were able to obtain agreement between the simulated and observed brightness tem-
peratures at frequencies between 325 and 664GHz using plausible particle habits. However, different habits gave the closest match in different channels and at different times, suggesting that it may be necessary to account for changes in particle shape, both horizontally and vertically within the cloud, in addition to changes in the size distribution. In contrast, for B895 we were unable to achieve satisfactory closure at 664GHz as the observed cloud signals were smaller than all of the simulations for

habits that were consistent with the in-situ observations. This was even the case for the 6HexagonalRosette particle that was specifically generated to have a mass-dimension relationship consistent with the in-situ observations for this flight as well as a shape representative of the typical CPI images. The reason for this failure is unknown, but one possibility is that horizontally aligned particles may enhance the lidar extinction and lead to an over-estimate of the particle sizes in the input to the radiative transfer model for this flight. In future, dual-polarized observations at off-nadir viewing angles may provide information on particle orientation and help to resolve such discrepancies.

The observed cloud signals were strongly dependent on ice water path. However, for a given ice water path they were also dependent on the ice water content distribution, with the thin, high IWC cloud during B939 leading to much larger signals than the thicker, lower IWC cloud during B895. This is likely to be primarily related to the particle size distribution, with more larger particles occurring at higher IWC. Whilst multi-frequency observations offer some sensitivity to particle size, retrievals are likely to remain challenging due to the limited information content and, particularly for cirrus clouds with relatively low IWP, the small signals at lower frequencies.

As well as showing sensitivity to ice water path and the distribution of ice mass within the cloud the simulations also demonstrate considerable sensitivity to the particle habits. This implies that accurate retrievals of cloud ice mass and particle size from passive millimetre and sub-millimetre observations will require the use of realistic habit models, and may require prior information on likely particle shapes to distinguish between changes in ice mass, particle size and particle habit.

The observed cloud signals were also compared to simulations using the SectorSnowflake habit that was found by Geer and Baordo (2014) to adequately reproduce satellite observations up to 183GHz on a global scale. However, we found it gave rather a poor match between 325 and 664GHz for these two case studies, with significant over-prediction of the cloud scattering. In part this is due to inconsistencies between the mass of the SectorSnowflake and the cloud particles present in these clouds, particularly for B939. However, even for B895 where the SectorSnowflake only overpredicts the mass by 60% there was still rather poor agreement with the observations at all the frequencies considered here.

In this study we have chosen to focus on nadir observations in order to give the closest match between the radiometer and lidar viewing directions, rather than considering the conical viewing geometry of ICI which has an incidence angle of 53°. Using nadir views also allows us to ignore polarisation differences that may be caused by the presence of horizontally aligned particles which are not currently included in the ARTS scattering database. The difference between nadir and slanted viewing geometries for horizontally aligned particles is discussed by Evans and Stephens (1995a), who show that the viewing angle dependence and polarisation dependence of the brightness temperature depression is strongly related. For randomly oriented particles we do not therefore expect significant differences between the two viewing geometries, so these results should be generally applicable to the satellite mission. The ARTS/RT4 solver used in this study is capable of simulating polarised brightness temperatures for oriented particles, and other solvers are available within ARTS (e.g. the 3D Monte-Carlo solver) which can use all elements of the scattering matrix, so in future it will be possible to fully evaluate the benefit of multi-frequency polarised observations for determining particle orientation.

The cirrus cases considered in this study represent the lower limits of cloud ice that can be observed by instruments such as ISMAR or ICI, and it would be desirable to extend the analysis to clouds with higher IWP. However, this is challenging due

to the difficulty of obtaining ground-truth observations of cloud profiles for input to the radiative transfer model. In particular, the lidar will not penetrate the full depth of thicker clouds so an alternative source of vertical profile information such as cloud radar is required. Cloud radar observations could also help to resolve the discrepancy between the observations and simulations for cases such as B895 by providing vertical profiles with significantly different sensitivity to the ice particle microphysics compared to the lidar. The general approach of combining active remote sensing and in-situ observations to constrain cloud properties for input to radiative transfer models could also be applied to other datasets, e.g. to cloud radar and CoSSIR observations from the TC4 or CRYSTAL FACE campaigns (Evans et al., 2012, 2005), although the exact methodology will vary depending on the available instrumentation. This could extend the validation of the radiative transfer models at sub-millimetre wavelengths to a greater range of meteorological conditions.

It would also be beneficial to obtain in-situ measurements more closely co-located with the remote sensing observations in order to reduce uncertainties due to evolution of the cloud with time. Such observations require campaigns using multiple aircraft and instrument packages, for example those available on the SAFIRE Falcon 20 and DLR HALO aircraft, although even with multiple aircraft it is not possible to simultaneously sample the full vertical extent of clouds. Obtaining suitable co-located active, passive and in-situ datasets should be a priority to allow the full exploitation of future satellite sensors such as ICI.

*Code and data availability.* The ARTS radiative transfer model can be downloaded from http://www.radiativetransfer.org. The single scattering database and associated tools are available from Zenodo (https://doi.org/10.5281/zenodo.1175572 (Ekelund et al., 2018) and https://doi.org/10.5281/zenodo.1175588 (Mendrok et al., 2018) respectively). FAAM data for flights B895 and B939 can be found on the NERC CEDA archive (http://catalogue.ceda.ac.uk/uuid/6ba397d6c8854da19bcced8ea588c1f9 and http://catalogue.ceda.ac.uk/uuid/dcc9dc73d8bc44caa51f5e8641f2c212)

*Competing interests.* The authors declare that they have no conflict of interest

*Acknowledgements.* ISMAR has been jointly funded by the Met Office and the European Space Agency under the "Cloud and Precipitation Airborne Radiometer" project. The authors would also like to thank the crew and personnel involved in the COSMICS and WINTEX-16 campaigns. The BAe-146 research aircraft is operated by Airtask and Avalon and managed by the Facility for Airborne Atmospheric Measurements (FAAM), which is jointly funded by the Met Office and Natural Environment Research Council (NERC). Non-core cloud microphysics measurements were funded by the NERC project "Cirrus Coupled Cloud-Radiation Experiment (CIRCCREX)" (grant references NE/K015133/1 and NE/K01515X/1) using instrumentation from The University of Manchester and the NERC National Centre for Atmospheric Science (NCAS) Atmospheric Measurement Facility (AMF).

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
