# Peer review of "Airborne validation of radiative transfer modelling of ice clouds at millimetre and sub-millimetre wavelengths"

_Atmospheric Measurement Techniques, 2018_

## Referee Comment (RC1) · Anonymous Referee #1 · 29 Oct 2018

Review comments for "Airborne validation of radiative transfer modelling of ice clouds at millimeter and sub-millimetre wavelengths" by Fox et al.

This work analyzed the pioneer airborne campaign data from ISMAR, a miniature of the ICI instrument that is planned to be launched to space in the near future, to demonstrate and explore the capabilities and difficulties of sub-mm passive microwave techniques would face. Other in-situ data (e.g., size and habit derived from CIP probes, and lidar retrieved PSD) are used together with a comprehensive radiative transfer model (RTM) called "ARTS" to make every effort to keep the retrieval results cross-instrument, cross-parameter and cross-frequency interconsistent. This approach adds

solidity to the results. The authors found that a great variety of ice particle habit could result in similar brightness temperature depression that are sometimes comparable to the observations while sometimes not. Same particle habits may achieve good agreement at certain channel frequency (e.g., 325 GHz) but not higher frequency (e.g., 664 GHz). Although one of the flight only flew through relatively thin cirrus cloud, it actually helped on identifying the detectability threshold of sub-mm technique on cloud ice water path/content, which is informative albeit the fact that RTMs had difficulties on reproducing this case (B895). Furthermore, the authors found out that one particle habit that worked well for mm-wave performs bad for sub-mm, indicating that we need to be very cautious when assuming one universal ice crystal shape across different spectrum regimes when it comes to the data assimilation in order to "truly" benefit the model and weather forecasting.

It is enjoyable reading this manuscript as the English and logic flow are smooth, and the results and methodology are nicely presented with many details and being strict at the same time. There are a few touches I hope the authors can take into consideration before final publication to make the discussion more in depth and open to future explorations at the same time.

(1) Please color the flight legs as a function of height so it's directly visible which heights the cloud information comes from.

(2) Other than to avoid H/V complication, why you particularly interested in only focusing on analyzing and comparing the near-nadir observations? Considering that the orientation information is most prominent at slant views, and considering that the ICI viewing geometry is designed to be conical (?, or am I have a wrong impression, I remember the viewing geometry is somewhat similar to GMI), near-nadir view story might not be so suggestive for the satellite-borne mission. Please do elaborate somewhere in the paper about what conclusions might be changed when we move to slantwise view? What are the potential difficulties that observations and RTM might face with the slantwise view, especially for other microphysical parameters that further perplex

the problem. Would multi-frequency V/H observations potentially helpful on resolving some of the issue, given that ARTS can now qualitatively capture the V/H through the Monte-Carlo method (i.e., not using the single scattering database, but using the 3D radiative transfer).

(3) It's not a very good assumption to assume the particle shape doesn't change over the whole flight leg, neither horizontally nor vertically. Also, it's natural somewhat to me to understand why 338 and 664 GHz cannot be achieved best-match at the same time, simply because they are sensitive to different size/shape of the particles which may likely co-exist in the vertical column at different altitudes. So I think you should discuss these possibilities that may likely happen in the real world and that partially account for the failure to match RTM simulations with observations.

(4) How do you deal with the antenna pattern (i.e., line function) for the ISMAR sensor? That may cause 1-2 K warm bias even if you have a perfect background atmosphere setting.

(5) B895 is really not an idea leg for this study because it's reaching the lower boundary of sub-mm sensitivity, as you also pointed out in your manuscript. So I would rather not put too much effort on match the B895 result – channel noise, imperfect background atmosphere, etc., all these factors can beat down the observed BT difference for this flight.

(6) I still don't get why the "smallPlateAggregate" produces the best match for IWC (Fig. 8), but not for IWP (Fig. 10)? It seems to me that the "problematic SectorSnowFlake" actually produces the best match for the BT difference – IWP relationship as shown in Fig. 10.

(7) In the discussion or conclusion section, please elaborate with a few sentences that whether your approach can be applied to previous campaigns, e.g., OLYMPEX with multi-frequency radar, CIP and CoSMIR? Further back in TC4 campaign, we have CoSSIR that is similar to ISMAR in some sense. If possible, using previous campaign

observations that carried out in different weather regimes might aid greatly on identify the sub-mm capability and RTM caveats/advantages.

---

## Referee Comment (RC2) · Anonymous Referee #2 · 19 Nov 2018

In this work, the authors perform a closure study trying to bring together observations in the submillimeter wavelength region taken by the ISMAR airborne demonstrator for ICI with radiative transfer simulations performed with the ARTS model and its accompanied single scattering database. The aim is to validate the radiative transfer setup, model, and the scattering database.

The manuscript nicely shows how difficult it is to perform such closure studies that try to match observations and models. This starts with the availability of appropriate instrumentation on suitable platforms and campaign setups that could provide all variables to constrain the atmosphere sufficiantly to accomplish the task. Evenmore, finding

suitable cases is not always possible.

Fox et al. present a work that faces some of these problems. By the instrument setup with having lidar for profile information only for low IWC and time shifted Nevzorov probe and insitu measurements on a limited number of levels, it is rather hard to constrain the atmospheric column. Especially for time shifts of up to one hour between remote sensing and insitu. As they mention themself, additional instrumentation like a radar and a second aircraft with the insitu probes flying more closely in time to the remote sensing suite, would have helped a lot.

Concerning the presented radiative transfer simulations with different particle types, I do not fully agree with the authors that the measured and simulated brightness temperatures are in the same range and represent the same variability. Strictly I would say, this is only the case for a few frequencies or parts of the flight legs. Although, they mention it is not within the scope of the study, one could consider varying the particle type along the legs or when flying in different altitudes as indicated by the paticle images. The extensive description of the particle habits in the database indicates the possibility of doing so. Here the shape information of the insitu probes could have been taken more into account.

In summary, the closure study did not succeed to find a match between observations and simulations over the whole measured spectrum and IWP range presented here.

I would recommend using the scan information provided by ISMAR if there are measurements under different angles during these flights. By this the study can be brought closer to ICI and could give information about orientation. It is too bad that interesting receiver channels did not work properly to perform a more in depth investigation of particle orientation.

448 +/- 1.4 Ghz is left out because the weighting function peaks very high in the atmosphere. Since the flights are very close to the clouds and high in the atmosphere, it might be worth taking them into account, eventhough the signal due to ice particles

scattering might be even smaller than in the other channels.

The influence of the surface in 243 GHz could be reduced by slanted simulations and observations. Over ocean it should be anyway possible to estimate the influence of the surface to a good degree. Especially in comparison between clear and cloudy sky, surface signal might not play a big role.

The derivation of the profiels of ice water content is not fully clear to me. I would appreciate of (average) profiles or time series of the IWC or IWP as utilized in the radiative transfer could be shown.

To my knowledge, there are coordinated flights of the BAe-146 with ISMAR on board with other aircraft like the HALO carrying water vapor lidar, radar and additional passive microwave instruments. Could these measurements help to constrain further the atmosphere and therefore the vertical distribution of ice water?

---

## Author Comment (AC1) · 21 Dec 2018

We thank the reviewer for their helpful comments. Our response to specific points is given below:

1. Please color the flight legs as a function of height so its directly visible which heights the cloud information comes from.

The aircraft tracks shown in Figure 1 in the manuscript are only for the above-cloud remote sensing runs. We will include additional plots (fig 1) showing the aircraft tracks during the in-cloud runs as a supplement to the revised manuscript, and update the text on p4 accordingly.

2. Other than to avoid H/V complication, why you particularly interested in only focusing on analyzing and comparing the near-nadir observations? Considering that the orientation information is most prominent at slant views, and considering that the ICI viewing geometry is designed to be conical (?, or am I have a wrong impression, I remember the viewing geometry is somewhat similar to GMI), near-nadir view story might not be so suggestive for the satellite-borne mission. Please do elaborate somewhere in the paper about what conclusions might be changed when we move to slantwise view? What are the potential difficulties that observations and RTM might face with the slantwise view, especially for other microphysical parameters that further perplex the problem. Would multi-frequency V/H observations potentially helpful on resolving some of the issue, given that ARTS can now qualitatively capture the V/H through the Monte-Carlo method (i.e., not using the single scattering database, but using the 3D radiative transfer).

ICI is indeed a conical scanner. As noted in the manuscript (p5, l20), our main reason for focusing on the near-nadir observations is to maintain consistency between the viewing direction of the radiometer and the nadir-pointing lidar. Although it would be possible to interpolate the lidar vertical profiles along the slant-path of the off-nadir radiometer observations the process could introduce additional uncertainties. Additionally, we would expect any impact from horizontally aligned particles would be more significant at off-nadir viewing angles, but the ARTS scattering database (and other available databases) currently only contain randomly oriented particles. Evans and Stephens [1995] discuss the effect of viewing direction on sensitivity to IWP, and show that for spherical particles it is approximately constant. We therefore expect little difference in the results for nadir and off-nadir viewing directions for randomly oriented particles.

Multi-frequency V/H observations are indeed expected to be helpful, particularly for detecting the presence of horizontally aligned particles [as discussed by Evans and Stephens, 1995, for example], and are included for the two window channels of ICI for this reason. Polarised brightness temperature can be simulated in ARTS with the RT4 solver used in this study, and the ARTS scattering database includes all required elements of the scattering matrix. However, as noted above the scattering database does not yet include horizontally aligned particles, and this limitation is common to other available databases.

We will include additional discussion in the conclusions of the revised manuscript about the applicability of our nadir results to the slantwise views relevant for ICI.

3. Its not a very good assumption to assume the particle shape doesn't change over the whole flight leg, neither horizontally nor vertically. Also, its natural somewhat to me to understand why 338 and 664 GHz cannot be achieved best-match at the same time, simply because they are sensitive to different size/shape of the particles which may likely co-exist in the vertical column at different altitudes. So I think you should discuss these possibilities that may likely happen in the real world and that partially account for the failure to match RTM simulations with observations.

We agree that particle shapes are likely to change both horizontally and vertically within the cloud, and that this may result in different single habits giving better matches at different frequencies or at different times; this is already discussed briefly in the text (p20). Using layered cloud models with different shapes in different altitude ranges could result in better agreement at all frequencies, but we would expect the brightness temperatures for a habit mixture to lie within the range of values predicted by the single habits. Given the difference in time and location between the in-situ and remote sensing observations and the limited microphysical information available from the lidar it is difficult to determine how suitable layered models could be derived based on the available observations. We will expand the discussion on this in the revised manuscript.

4. How do you deal with the antenna pattern (i.e., line function) for the ISMAR sensor? That may cause 1-2 K warm bias even if you have a perfect background atmosphere setting.

We do not account for the antenna pattern and assume an idealised pencil beam. For the relatively narrow ISMAR beamwidths (less than 4 degrees FWHM) and nadir viewing geometry, clear sky simulations suggest that the impact of the main beam width will be significantly less than 0.1K. For off-nadir views the impact of the antenna pattern is much greater (order 1K) and would need to be accounted for in the simulations. We will include this information in section 4 of the revised manuscript.

5. B895 is really not an idea leg for this study because its reaching the lower boundary of sub-mm sensitivity, as you also pointed out in your manuscript. So I would rather not put too much effort on match the B895 result channel noise, imperfect background atmosphere, etc., all these factors can beat down the observed BT difference for this flight.

Although it is true that B895 is approaching the limits of sub-mm sensitivity we think that it is notable that none of the habits with sufficient ice mass for a given particle size are capable of simulating the low sensitivity of the 664GHz observations.

6. I still dont get why the smallPlateAggregate produces the best match for IWC (Fig. 8), but not for IWP (Fig. 10)? It seems to me that the problematic SectorSnowFlake actually produces the best match for the BT difference IWP relationship as shown in Fig. 10.

We do not fully understand this comment. Fig. 8 is a time-series of brightness temperatures and does not refer to IWC. The smallPlateAggregate produces a good match for IWC in figure 5, which simply implies that it has a mass-dimension relationship that is consistent with the in-situ PSDs and bulk IWC measurements. The fact that it gives a poor match to the brightness temperatures in fig. 10 shows that it does not have consistent scattering properties. We do not agree that the SectorSnowFlake provides a good match to the BT difference - IWP relationship in Fig 10 except perhaps for a few outlying points at 448 and 664GHz. This is further demonstrated in Table 3, where it has the largest biases and RMS differences of all the habits considered.

7. In the discussion or conclusion section, please elaborate with a few sentences that whether your approach can be applied to previous campaigns, e.g., OLYMPEX with multi-frequency radar, CIP and CoSMIR? Further back in TC4 campaign, we have CoSSIR that is similar to ISMAR in some sense. If possible, using previous campaign observations that carried out in different weather regimes might aid greatly on identify the sub-mm capability and RTM caveats/advantages.

The general approach used here of combining in-situ observations with active remote sensing to constrain the cloud properties for input to radiative transfer models could certainly be applied to other datasets, although the details will vary depending on the available instrumentation (radar, lidar etc). Indeed, similar techniques have already been applied to CoSMIR [Olson et al., 2016], although the relatively low frequency channels (up to 183GHz) make it less applicable to ICI. Revisiting the CoSSIR observations from TC4 and CRYSTAL FACE could be interesting, although the existing retrieval studies [Evans et al., 2012, 2005] are, to a certain extent, a rather indirect validation of the radiative transfer models. We will mention the possibility of using CoSSIR observations in the conclusions of our revised manuscript.

**References**

- K. F. Evans, J. R. Wang, D. O'C Starr, G. Heymsfield, L. Li, L. Tian, R. P. Lawson, A. J. Heymsfield, and A. Bansemer. Ice hydrometeor profile retrieval algorithm for high-frequency microwave radiometers: application to the CoSSIR instrument during TC4. *Atmospheric Measurement Techniques*, 5(9):2277–2306, 2012. doi: 10.5194/amt-5-2277-2012. URL http://www.atmos-meas-tech.net/5/2277/2012/.
- K. Franklin Evans, James R. Wang, Paul E. Racette, Gerald Heymsfield, and Lihua Li. Ice cloud retrievals and analysis with the compact scanning submillimeter imaging radiometer and the cloud radar system during crystal face. *Journal of Applied Meteorology*, 44(6):839–859, 2005. doi: 10.1175/JAM2250.1. URL http://dx.doi.org/10.1175/JAM2250.1.
- KF Evans and GL Stephens. Microwave radiative transfer through clouds composed of realistically shaped ice crystals. part ii: remote sensing of ice clouds. *Journal of the Atmospheric Sciences*, 52(11):2058–2072, 1995. doi: 10.1175/1520-0469(1995)052j2058:MRTTCCj2.0.CO;2.
- William S. Olson, Lin Tian, Mircea Grecu, Kwo-Sen Kuo, Benjamin T. Johnson, Andrew J. Heymsfield, Aaron Bansemer, Gerald M. Heymsfield, James R. Wang, and Robert Meneghini. The microwave radiative properties of falling snow derived from nonspherical ice particle models. part ii: Initial testing using radar, radiometer and in situ observations. *Journal of Applied Meteorology and Climatology*, 55(3):709–722, 2016. doi: 10.1175/JAMC-D-15-0131.1. URL https://doi.org/10.1175/JAMC-D-15-0131.1.

Figure 1:  $10.8\mu$ m polar orbiting satellite images and aircraft tracks during in-situ sampling runs for B895 (top) and B939 (bottom). The satellite, overpass time, run altitude and run times are indicated on the plots.

---

## Author Comment (AC2) · 21 Dec 2018

We thank the reviewer for their comments, and our responses are given below.

*Concerning the presented radiative transfer simulations with different particle types, I do not fully agree with the authors that the measured and simulated brightness temperatures are in the same range and represent the same variability. Strictly I would say, this is only the case for a few frequencies or parts of the flight legs. Although, they mention it is not within the scope of the study, one could consider varying the particle type along the legs or when flying in different altitudes as indicated by the particle images. The extensive description of the particle habits in the database indicates the possibility of doing so. Here the shape information of the insitu probes could have been taken more into account.*

*In summary, the closure study did not succeed to find a match between observations and simulations over the whole measured spectrum and IWP range presented here.*

We agree that no single particle habit is capable of reproducing all the observations at different frequencies and locations, and this may partly be caused by variations in particle habit spatially and through the depth of the cloud, combined with the different sensitivities of the various channels considered. However, as discussed in our response to reviewer 1 we do not think that the in-situ observations are representative enough to generate sufficiently accurate layered cloud models. Rather our approach is to use the in-situ observations to eliminate implausible particle habits (through the mass-dimension relationship), and consider whether the remaining shapes are capable of simulating the observed brightness temperatures. However, even with the wide range of particles available in the ARTS database this is not always possible and our attempt to generate a rosette-type particle consistent with the in-situ observations for B895 was not able to reproduce the observations at 664GHz. We will expand the discussion around this in the revised manuscript.

*I would recommend using the scan information provided by ISMAR if there are measurements under different angles during these flights. By this the study can be brought closer to ICI and could give information about orientation. It is too bad that interesting receiver channels did not work properly to perform a more in depth investigation of particle orientation.*

Please see our response to reviewer 1 regarding using off-nadir scan angles closer to the viewing geometry of ICI. We have focused on the nadir views to be consistent with the lidar viewing direction and to reduce the impact of horizontally aligned particles which are not currently present in the ARTS scattering database. Since the main difference between nadir and off-nadir views is expected to be polarisation effects from oriented particles, detailed consideration of off-nadir views is beyond the capabilities of the current simulations.

*448 +/- 1.4 Ghz is left out because the weighting function peaks very high in the atmosphere. Since the flights are very close to the clouds and high in the atmosphere, it might be worth taking them into account, eventhough the signal due to ice particles scattering might be even smaller than in the other channels.*

We do not think that including the 448±1.4 GHz results would add anything to the paper as the cloud signals are very small - the simulated brightness temperature depressions for this channel are generally below 2K except for the SectorSnowflake during the initial parts of B939. This channel is also very sensitive to the upper troposhperic water vapour profile which is not well constrained by the available 183GHz observations, leading to larger uncertainties in the clear-sky simulations that are used as a baseline to determine the cloud-induced signals. The plots for the two flights for this channel are shown in figs 1 and 2 for reference; the large scatter in the observations is probably due to uncertainties in the water vapour profile.

*The influence of the surface in 243 GHz could be reduced by slanted simulations and observations. Over ocean it should be anyway possible to estimate the influence of the surface to a good degree. Especially in comparison between clear and cloudy sky, surface signal might not play a big role.*

At 243GHz, Prigent et al. [2016] showed that the mean error in surface upwelling brightness temperature using ERA-Interim atmospheric and surface fields and the FASTEM ocean emissivity model is ∼4K. Since the clear-sky transmission at 243GHz is ∼0.5 at nadir (and ∼0.36 at 50 degrees) this could introduce a significant bias in the clear-sky simulations that is of similar magnitude to the expected cloud signals. Although the surface signal may not be significant when calculating simulated cloud-induced brightness temperature depressions (i.e. cloudy simulation - clear sky simulation), the calculation of the observed cloud-induced brightness temperature depression (i.e. cloudy observation - clear sky simulation) requires unbiased clear-sky simulations for the comparison to be representative. For cases over land the surface emissivity may also be rather variable and is not well constrained. We therefore do not feel that the 243GHz results are worth including in the paper. Nevertheless, we include the relevant plots in fig 1 for reference, and it can be seen that the scatter in the observations hides any underlying variations in the cloud signal in response to changing IWP.

*The derivation of the profiels of ice water content is not fully clear to me. I would appreciate of (average) profiles or time series of the IWC or IWP as utilized in the radiative transfer could be shown.*

The profiles of ice-water content (and hence the integrated ice water path) are estimated from the profiles of lidar extinction using flight-dependent empirical relationships derived from the in-situ observations as described briefly on p15 l16. Note that this ice mass profile is not directly used in the radiative transfer simulations which instead use lidar-derived profiles of the second moment of the PSD to fix the size distribution. However, they are used to calculate the IWP used to plot Fig 9 and Fig 10. We will expand the description of the method and include plots of the lidar-derived IWC (shown below in fig 2) in the revised manuscript.

[Figure]

Figure 1: Cloud-induced brightness temperature differences as a function of time and IWP at 243 and 448±1.4 GHz for B895 (top) and B939 (bottom)

*To my knowledge, there are coordinated flights of the BAe-146 with ISMAR on board with other aircraft like the HALO carrying water vapor lidar, radar and additional passive microwave instruments. Could these measurements help to constrain further the atmosphere and therefore the vertical distribution of ice water?*

A co-ordinated flight between the FAAM BAe-146, HALO and the Safire Falcon was performed in October 2016 bringing together ISMAR, multi-frequency cloud radar, lidar, additional passive microwave instruments and in-situ observations of a deeper cloud system than the cases used in this study. This flight will be the subject of future studies.

**References**

C. Prigent, F. Aires, D. Wang, S. Fox, and C. Harlow. Sea-surface emissivity parametrization from microwaves to millimetre waves. *Quarterly Journal of the Royal Meteorological Society*, pages n/a–n/a, 2016. ISSN 1477-870X. doi: 10.1002/qj.2953. URL http://dx.doi.org/10.1002/qj.2953.

[Figure]

Figure 2: Lidar-derived ice water content profiles for B895 (top) and B939 (bottom)

---

## Referee Report (RR1)

Stuart Fox et al. present in their manuscript „Airborne validation of radiative transfer modelling of ice clouds at millimetre and sub-millimetre wavelengths" a closure study between ISMAR observations and forward simulations with ARTS radiative transfer model by using a single scattering database. The atmospheric setup for the forward simulation is based on adjusted numerical model output and hydrometeor profiles derived from lidar and in-situ probes observations.

Although no full closure can be achieved for the presented frequencies, the study gives a valuable contribution to the fields  of remote sensing of ice cloud proberties in the millimeter and submillimeter wavelength region from aircraft and satellite.

The answers the authors gave to the comments in the first review are sufficient and clarified open questions. I highly appreciate that they provided additional figures.

For this review I have only two open question  and some technical comments.

It is mentioned that the study has been done as well for the MARSS frequency 183GHz but no results are presented.

p.24 l.19: I think th point that only nadir observations are used because there you have the closest match to the lidar doesn't count? The lidar is only used to derive the ice water  content. Once the the atmosphere is set up, the simulations can be performed for any observation angle and therefore compared to the ISMAR observations.

To my imagination the sentences are sometimes rather long.

To me it is a slight inconsistency writing „millimetre and sub-millimetre" and „millimetre/submillimetre".

p24. l.21 + l.23: horizintally-aligned and viewing-angle without hyphen

---

## Author Response (AR2)

**1 Response to referee #1**

We thank the referee for their comments. It is true that RT4 is not capable of simulating all elements of the Stokes vector, and we have modified the wording on page 24 (l30-33) to address this.

**2 Response to referee #2**

We thank the referee for their comments, and our responses are as follows:

- Results are only presented for 325 GHz and above due to the low sensitivity to thin cirrus clouds at the lower frequencies. However, we have still used the 183 GHz observations to constrain the atmospheric water vapour profile. This has been clarified on p4 (l29-31)

- It is true that the simulations could be performed for a 1D atmosphere at any observation angle, but we still believe that the lack of horizontally aligned particles in the ARTS scattering database prevents direct comparisons.

- We have replaced all incidences of "millimetre/submillimetre" with "millimetre and sub-millimetre"

- We have removed the hyphens on p24

**Airborne validation of radiative transfer modelling of ice clouds at millimetre and sub-millimetre wavelengths**

Stuart Fox[1], Jana Mendrok[2], Patrick Eriksson[2], Robin Ekelund[2], Sebastian J. O'Shea[3], Keith N. Bower[3], Anthony J. Baran[1,4], R. Chawn Harlow[1], and Juliet C. Pickering[5]

[1]Met Office, FitzRoy Road, Exeter, UK, EX1 3PB
[2]Department of Space, Earth and Environment, Chalmers University of Technology, 41296 Gothenburg, Sweden
[3]Centre for Atmospheric Science, School of Earth and Environmental Science, University of Manchester, Manchester, UK, M13 9PL
[4]School of Physics, Astronomy and Mathematics, University of Hertfordshire, Hatfield, UK, AL10 9AB
[5]Space and Atmospheric Physics Group, Imperial College London, London, UK

*Correspondence to:* S. Fox (stuart.fox@metoffice.gov.uk)

**Abstract.** The next generation of European polar orbiting weather satellites will carry a novel instrument, the Ice Cloud Imager (ICI), which uses passive observations between 183 and 664GHz to make daily global observations of cloud ice. Successful use of these observations requires accurate modelling of cloud ice scattering, and this study uses airborne observations from two flights of the Facility for Airborne Atmospheric Measurements (FAAM) BAe-146 research aircraft to validate radiative trans-

5 fer simulations of cirrus clouds at frequencies between 325 and 664GHz using the Atmospheric Radiative Transfer Simulator (ARTS) and a state-of-the-art database of cloud ice optical properties. Particular care is taken to ensure that the inputs to the radiative transfer model are representative of the true atmospheric state by combining both remote-sensing and in-situ observations of the same clouds to create realistic vertical profiles of cloud properties that are consistent with both observed particle size distributions and bulk ice mass. The simulations are compared to measurements from the International

10 Sub-Millimetre Airborne Radiometer (ISMAR), which is an airborne demonstrator for ICI. It is shown that whilst they are generally able to reproduce the observed cloud signals, for a given ice water path (IWP) there is considerable sensitivity to the cloud microphysics including the distribution of ice mass within the cloud and the ice particle habit. Accurate retrievals from ICI will therefore require realistic representations of cloud microphysical properties.

*Copyright statement.* The works published in this journal are distributed under the Creative Commons Attribution 4.0 License. This li-

15 cence does not affect the Crown copyright work, which is re-usable under the Open Government Licence (OGL). The Creative Commons Attribution 4.0 License and the OGL are interoperable and do not conflict with, reduce or limit each other.

© Crown copyright 2018

[revised manuscript text omitted]